# Computational models of category-selective brain regions enable high-throughput tests of selectivity

N. Apurva Ratan Murty [1,2,3,5 ✉], Pouya Bashivan [4,5], Alex Abate[1], James J. DiCarlo [1,2,3] & Nancy Kanwisher [1,2,3]

Cortical regions apparently selective to faces, places, and bodies have provided important evidence for domain-specific theories of human cognition, development, and evolution. But claims of category selectivity are not quantitatively precise and remain vulnerable to empirical refutation. Here we develop artificial neural network-based encoding models that accurately predict the response to novel images in the fusiform face area, parahippocampal place area, and extrastriate body area, outperforming descriptive models and experts. We use these models to subject claims of category selectivity to strong tests, by screening for and synthesizing images predicted to produce high responses. We find that these high-response-predicted images are all unambiguous members of the hypothesized preferred category for each region. These results provide accurate, image-computable encoding models of each category-selective region, strengthen evidence for domain specificity in the brain, and point the way for future research characterizing the functional organization of the brain with unprecedented computational precision.

[1] Department of Brain and Cognitive Sciences, Massachusetts Institute of Technology, Cambridge, MA, USA. [2] McGovern Institute for Brain Research, Massachusetts Institute of Technology, Cambridge, MA, USA. [3] The Center for Brains, Minds and Machines, Massachusetts Institute of Technology, Cambridge, MA, USA. [4] Department of Physiology, McGill University, Montréal, QC, Canada. [5] These authors contributed equally: N. Apurva Ratan Murty, Pouya Bashivan. ✉email: ratan@mit.edu

The discovery of cortical regions apparently specialized for the perception of faces[1], places[2], and bodies[3] has not only enriched our understanding of the functional organization of the human brain, but energized longstanding debates about the structure, evolution, and development of the human mind. After all, faces, bodies, and places are highly meaningful stimuli at the core of two human abilities: engaging in complex social interactions with other people, and finding our way around in the world. Extensive research has further shown that these abilities follow distinct developmental trajectories in infancy[4,5] and are subserved by different representations and computations in adults. Thus, a theoretically rich account of category-selective regions in the brain requires understanding the meaning and significance of faces, places, and bodies to humans, and how the perception of these stimuli might be tailored to their post-perceptual uses in social cognition and navigation[6,7].

On the other hand, the intuitive definitions of faces, places, and bodies that figure in theories of cognitive architecture, development, and evolution are incomplete as characterizations of neural responses. First, they are not image computable, instead requiring a human in the loop to ascertain what counts as a face, place or body. Second, they provide no quantitative account of reliable differences in each region's response to images either within, or outside, its hypothesized preferred category. Finally, although considerable evidence supports the hypothesized category selectivities of the fusiform face area (FFA)[8,9], extrastriate body area[3,10,11] (EBA), and the parahippocampal place area (PPA)[2,12–14], each hypothesis remains vulnerable to refutation. Despite the hundreds of stimuli whose responses have been reported in the literature on each region, a vast space of images remains untested. If any image not from the preferred category is found in the future to maximally drive the region, the claim of category selectivity of that region will be seriously challenged. That is, a real possibility exists that the claimed category selectivity of the FFA, PPA, or EBA could turn out to be false. Here we tackle all three problems by developing image-computable encoding models that accurately predict the response of each region to new images and generalize across participants. We then cycle back to use these models to conduct the strongest tests to date of the hypothesized category selectivity of these regions.

Our work is made possible by recent advances in deep convolutional artificial neural networks (ANNs), based loosely on the hierarchical architecture and repeated computational motifs observed in the primate visual system[15,16]. These networks now approach human-level performance on object recognition benchmarks, providing the first computationally explicit hypotheses of how these tasks might be accomplished by the brain. Further, the internal representations developed at different processing stages within these ANNs mirror the hierarchical organization of the visual cortex[17–19], and activations in these networks can be linearly combined to accurately predict the observed response to previously unseen images at different stages of the visual processing hierarchy[20–23]. For these reasons, specific ANNs are now considered our most quantitatively accurate computational models of visual processing in the primate ventral visual stream[21]. However, it remains unclear whether or how the understanding provided by these models engages with previous theories of visual processing in the brain[24], or whether they even represent any significant advance in our understanding beyond what is already known from decades of published work on these regions.

In this work, we addressed these questions by collecting high-quality event-related functional MRI (fMRI) responses in the FFA, PPA, and EBA and screening a large number of ANN-based models of the ventral stream for their ability to predict observed responses in each region. Using prediction as one metric of understanding, we further tested whether these models outperform experts on the human ventral visual pathway at predicting the fMRI responses to novel images. Finally, we adapted recent machine learning methods to identify stimuli that maximally or differentially drive single neurons in the macaque visual system[25–27], to identify optimal stimuli for the FFA, PPA, and EBA. This method enables us to turbo-charge the search for counterevidence to the claimed selectivity of the FFA, PPA, and EBA, thereby conducting strong tests of longstanding hypotheses about the category selectivity of each region.

We show here that our models accurately predict the response of each region to images, even outperforming predictions from prior descriptive models and experts in the fields. This enables us to use these models to screen millions of images and synthesize naturalistic stimuli predicted to produce the highest response in each region. All the high-response predicted images for each region are obvious exemplars of the hypothesized preferred category for that region, supporting the claimed category selectivity of each region.

## Results
We scanned four participants with fMRI to first localize the FFA, PPA, and EBA in each participant individually using a standard dynamic localizer[28–30], and then recorded event-related fMRI responses in each of these functionally-defined regions of interest (fROIs) to a diverse set of 185 naturalistic stimuli. Each of the 185 images was presented at least 20 times to each participant over four scanning sessions (~10 h scanning in each of $N = 4$ participants), producing highly reliable responses of regions and voxels in the ventral pathway to these stimuli (Figs. S1, S2).

**ANN models of the ventral stream accurately predict responses to the FFA, PPA, and EBA.** How well do computational models of the ventral stream predict the observed response to natural stimuli in the FFA, PPA, and EBA? To find out, we modeled the average response across participants of each of six fROIs (left and right FFA, PPA, and EBA) to 185 natural images using a regression-based model-to-brain alignment approach[20,21,23,26,31,32] (Fig. 1). Specifically, given a model, we established a linear mapping between a selected layer of the model and the activation of each brain region. To determine the weights of that linear mapping, we used the brain region's measured responses (mean within a fROI, averaged over participants) to a subset of the stimuli (randomly selected) (Fig. 2). That is, the response of each ROI is modeled as a fixed, weighted sum of ANN features. We then tested the accuracy of this model at predicting responses to completely held-out stimuli (aka. cross-validation, scored as the Pearson correlation of the predicted vs. the observed responses on those held-out stimuli, see section on Encoding Models in Methods). Using this approach we screened a range of models on their ability to predict the observed responses across the different fROIs (i.e., integrative benchmarking[21,33]). These models include pixel and Gabor wavelet-based V1 models which extract low-level features as well as several popular ANNs, considered the leading models of the primate ventral visual stream[20,21,26] ($N = 60$ models, Table S1).

The results from the broad model screen are presented for all fROIs in Fig. S3, which shows that several ANNs have high cross-validated neural predictivity (i.e., above 0.8). Comparison of predictivity scores across models also reveal several notable trends. First, deep ANN models of the ventral stream surpass simple pixel and V1-level models. Second, prediction scores were higher for deeper ANN models (or recurrent models like CORnet-R[34] and CORnet-S) than shallower models (with no recurrence) like CORnet-Z or even AlexNet. Third, models trained on broad stimulus categories (like ImageNet and Places,

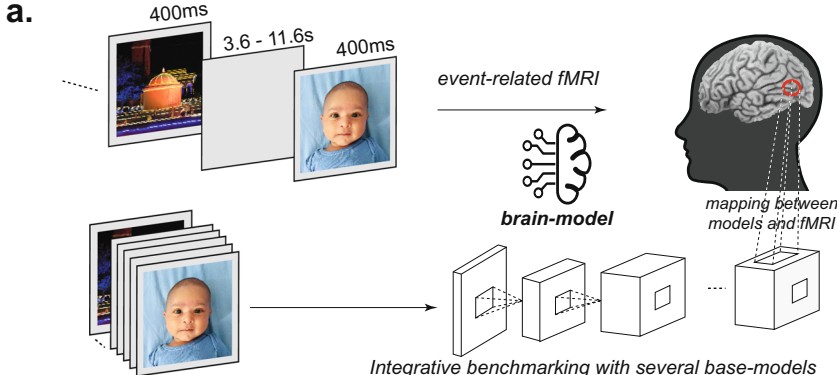

**Fig. 1 Outline of experiment design, fROI modeling. a** Experiment design and fROI modeling. BOLD responses to each image were collected using an event-related fMRI experiment (*top*), and the functional ROIs (FFA, EBA, PPA) were defined in each subject using an independent functional localizer (schematically outlined on the brain in red). The same images were then fed to several candidate image-computable models (base-models, below). Using a subset of stimuli, we learn a linear mapping (right) from each model to each brain region (see Methods) to build a brain model for each region based on each CNN model. The brain-models for each region were then evaluated based on the ability to predict the responses to held-out images not used in the mapping (integrative benchmarking). Note that images here are used only for illustrative purposes and are not the actual images used in the experiment. Permission was obtained from the parents of the person in the photograph to publish the image used in this figure.

which contain diverse naturalistic images) better predict neural responses than models trained on a specific domain of stimuli (like faces). This conclusion is based on comparing the neural predictivity for models with the same architecture backbone (say Resnet-50), but with synaptic weights trained on object categorization using different stimulus datasets (like IMAGENET[35], Places365[36], or VGGfaces2[37]). And finally, we find that models that are trained (see ResNet-50 random, Fig. S3) are much more accurate than models that have randomly initialized synaptic weights (see ResNet-50-random, Fig. S2). Figure 2a–c shows the striking correlation between the predicted and observed response to each image, separately for each hemisphere of the FFA, EBA, and the PPA, for one of the best ANN models—Resnet50-V1. Note that these correlations have not been corrected by the data reliability (though see Methods under Encoding Models). Together, the results from Fig. 2 and Fig. S2 show that the models for each fROI based on a trained ResNet50 are able to predict the observed average response to previously unseen stimuli with high accuracy (consistent with ref. [38]).

These encoding models for the FFA, PPA, and the EBA would be most useful if they also generalized to predict the observed responses from entirely new subjects. To find out how well our current models do this, we next built models based on pooling data from three of the participants and evaluating how well this model generalizes to new stimuli in the held-out participant (cross-validation across both subjects and images). Here too we found that models predict the observed responses in the held-out subject, with average correlations between predicted and observed responses for all fROIs above $R = 0.78$, (mean ± s.e.m across fROIs $0.82 ± 0.01$, each $P < 0.00005$, Fig. S4). How many participants do we need to obtain a good model for a given fROI? And do our methods work well even when models are built from a single participant, without first averaging responses across three or more participants? To find out we measured predictive accuracy when models were built based on a single participant's responses to 90% of stimuli and tested on the held-out 10% stimuli for the same participant. Here too the correlations between predicted and observed responses were high (each $R > 0.78$, mean ± s.e.m across fROIs $0.83 ± 0.01$ for all fROIs, each $P < 0.00005$, Fig. S4). Finally, we asked whether the models built from individual subjects generalize to other subjects by measuring

the predictive accuracy of a model built from one subject on another individual's responses to unseen images. Here again predictive accuracy remained quite high, with the correlations between predicted and observed responses $R > 0.76$ for all fROIs (mean ± s.e.m across fROIs $0.79 ± 0.01$, each $P < 0.00005$, see Fig. S4).

The results described so far indicate that the encoding model for each fROI generalizes across participants, but they do not yet address the grain of predictions across stimuli. Do these models predict the responses to individual images, over and above predictions based on their category membership? The fact that the high predictive accuracy of the pooled model in Fig. 2a–c was also observed in a separate analysis of stimuli both within the hypothesized preferred category, and outside the preferred category (Fig. 2a blue and pink dots, correlations shown as insets) already provides some evidence that they do. We further tested this question in two ways. First, we randomly shuffled the image labels ($N = 100$ iterations) but only within the face, body, scene, and object categories (within-category shuffled control) and estimated the correlation between this shuffled order and the predicted activation. This correlation was significantly lower than the unshuffled correlation between the predicted and observed response (Fig. 2e, $P = 0.03$, Wilcoxon sign-rank test), indicating that models explained unique image-level variance over and above the different mean responses to each category. Second, we estimated the degree to which the models could predict the observed response to individual stimuli within each of the four stimulus categories. If the models predict no image-level variance, the correlation within each category should be 0 (Fig. 2f, dotted line). Instead, each of these correlations were significantly greater than 0 for all fROIs (Fig. 2f, mean ± s.e.m across fROIs and categories $0.56 ± 0.03$, $P = 1.19 \times 10^{-7}$ Wilcoxon sign-rank test). Further, within-category predictivity was higher for the preferred than nonpreferred categories for each region, as expected from the plausible hypothesis that these regions are more sensitive to variation across exemplars within their preferred category than variations across exemplars within nonpreferred categories. Taken together, these results show that ANN models of the ventral stream can predict the response to images in the FFA, PPA, and the EBA with very high accuracy. Further analyses on a trained ResNet50 show that the predictive accuracy of the models remain high even when tested on individual stimuli within

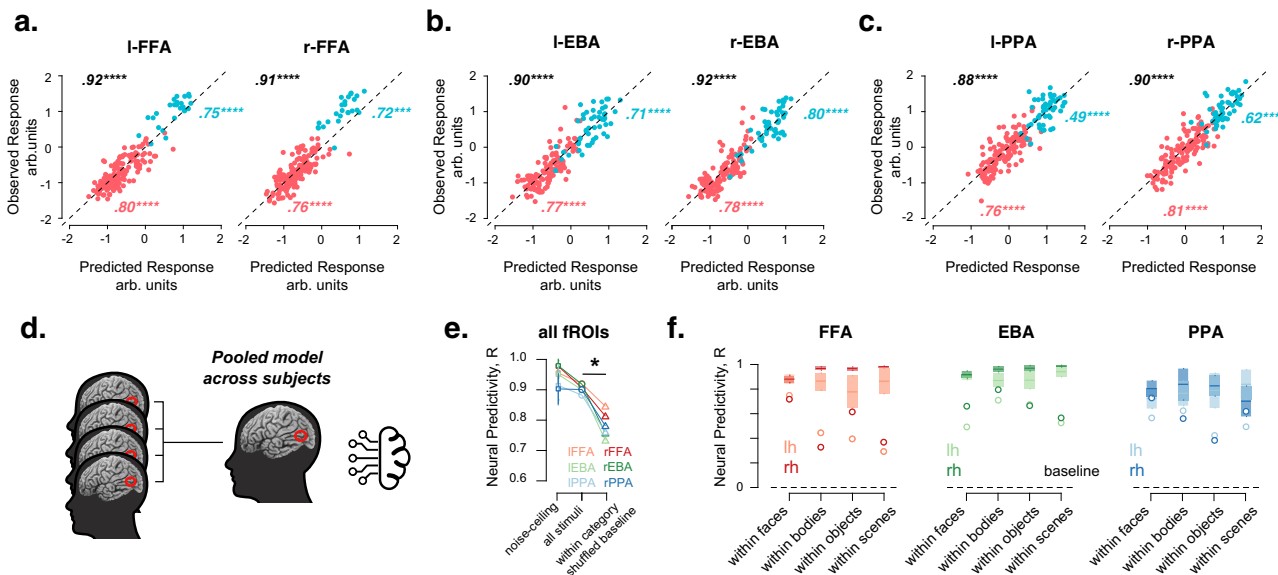

**Fig. 2 Models accurately predict image-level responses of the FFA, PPA, and EBA and generalize across subjects. a** Scatterplots show the cross-validated predicted response (*x*-axis) v/s observed response (*y*-axis) for each of the 185 stimuli for the left and right FFA (left and right columns, respectively). Numbers indicate correlation between the predicted and observed data across all stimuli (black text), for stimuli within the hypothesized preferred category (blue dots and text) and stimuli not within the hypothesized preferred category (pink dots and text). The dotted line denotes the x = y line and **** is $P < 0.00005$. Source data are provided as a Source Data file. **b** Same as **a** but for left and right EBA. Source data are provided as a Source Data file. **c** Same as **a** but for left and right PPA. Source data are provided as a Source Data file. **d** Schematic describing how the data were pooled across subjects. Data were averaged within each fROI (schematized as a red circle) in every subject and pooled across subjects and the computational models were built to predict the pooled responses for previously unseen images across participants. **e** Computational models predict the responses to individual stimuli within each of the fROIs (circles) significantly better than the within-category image shuffled baseline controls (triangles). Ticks indicate the mean and the errorbars indicate the std over samples. $N = 6$ fROIs. * is $P = 0.03$, two-sided Wilcoxon signed-rank test across fROIs, demonstrating predictive accuracy as a function of not just category but individual images. Source data are provided as a Source Data file. **f** Computational models predict responses to individual stimuli even for stimuli within each of the categories. Each circle indicates the model performance for the FFA (left column), EBA (middle column), and the PPA (right column), the dashed line at 0 indicates the expected baseline performance if models did not predict any image-grained variance within the categories. The shaded regions indicate the estimated noise-ceiling of the data. Source data are provided as a Source Data file.

categories and even across participants. Testing predictions within categories also exposed a larger gap between the model predictions and responses, which indicates room for further model-development efforts.

**ANN models of the ventral stream also predict voxel-wise and population-level representational dissimilarities across images in the FFA, PPA, and EBA in individual subjects.** The previous analyses evaluate the ability of models to predict the pooled response averaged across voxels within each category-selective fROI in an effort to build models that generalize across subjects and directly interface with the bulk of prior experimental literature on these regions. But of course information in these regions is coded by the distributed pattern of response across the neural population. So, how well do these models predict the voxel-wise responses (similar to other modeling studies[22,23,32,38–40]) in each region? And how correlated are the voxel-wise metrics of predictivity to the pooled population metrics, across models? To find out, we screened all 60 models as before, but this time on each individual voxel in every participant, and to the patterns of responses across voxels within a region. Specifically, we calculated the correlation between model predictions and the observed neural response for all the models using two population-level metrics: (i) a neural predictivity metric based on building models for every voxel and measuring the median correlation between the observed and predicted responses across voxels, and (ii) a population-level Representational Dissimilarity Matrix (RDM)-based metric, which was computed by comparing the observed population RDM within each fROI with the model-predicted

RDM based on transforming the voxel-wise predictions into RDMs. The neural predictivity scores for all 60 models tested on both population-level metrics are shown in Fig. S5. We observed a striking match between the fROI mean and the voxel-wise scores (Spearman $R = 0.99$, $P < 0.00001$, $N = 60$ models). Note, however, that the neural predictivity scores for the mean-fROI responses were in general higher, presumably because averaging across voxels increased SNR. It is also clear that the models that best predict the mean responses within individual ROIs, also best predict the voxel-wise responses and the RDMs (Fig. S5, and see Fig. S6 for the observed and predicted RDMs for a few representative models) in individual subjects. Together these results show the success of our encoding models at predicting the voxel-wise responses as well as population-based RDMs of these fROIs with high accuracy, and that the pooled and population-level metrics are highly correlated for these data.

**ANN models of the ventral stream have higher predictive accuracy than category-based descriptive models and domain experts.** As we have demonstrated, our encoding models for the FFA, EBA, and the PPA based on a trained ResNet50 are highly accurate at predicting the observed responses to individual images, even within the preferred category. This finding indicates that the models are capturing something beyond a binary distinction between the preferred versus nonpreferred categories. But how much better are these models compared to the previous descriptive characterizations used for these regions? That is, are the models capturing a graded version of category selectivity, much like human intuitions of graded category membership[41,42]? To find out, we

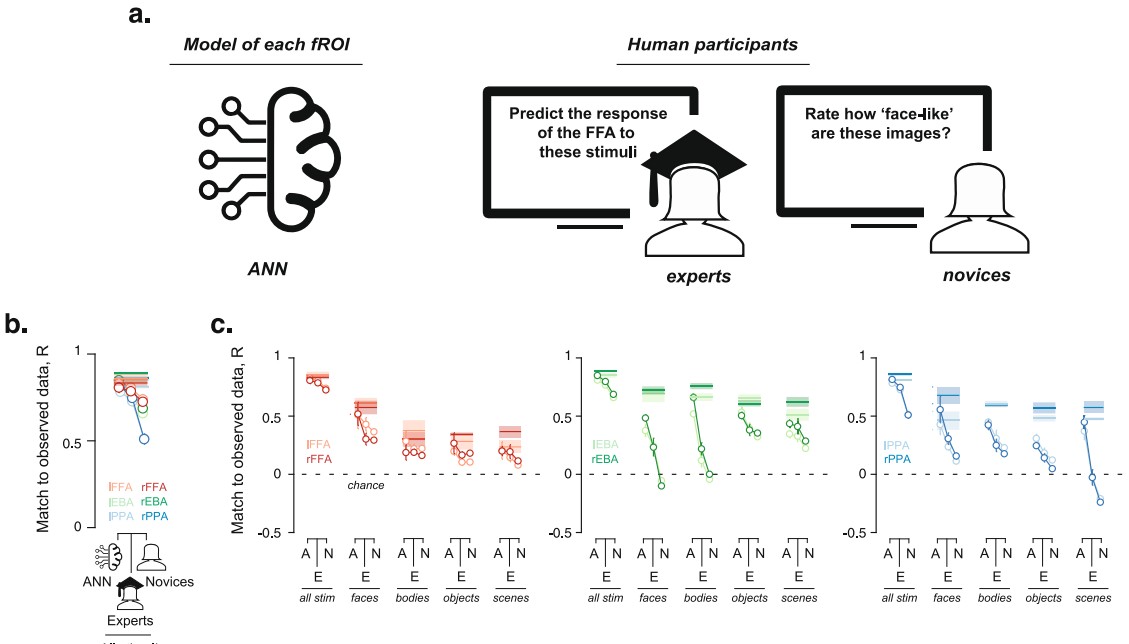

**Fig. 3 Model-based predictions compared to novice participants and experts in the field. a** Schematic of the comparison between neural network models for each fROI (left) and human participants who were either Professors with several published papers on the human ventral visual cortex (experts) or participants from a crowd-sourced experimental platform (novices), and **b** Performance of the ANN-based encoding models (left column), experts (middle column), and novice participants (right column) at predicting the observed responses to stimuli. The connected lines indicate each fROI. For the behavioral data, each dot indicates the mean prediction and the errorbars indicate the s.e.m. across participants. Source data are provided as a Source Data file. **c** Comparison between ANN models (A), experts (E), and novice (N) participants at predicting the observed responses at the grain of individual stimuli within each of the four categories of stimuli used in the experiment for the FFA (left column), the EBA (middle column), and the PPA (right column). For the behavioral data, each dot indicates the mean and the errorbars indicate the s.e.m. across participants. The line and shaded regions indicate the ceiling for how good the models can be, based on the reliability of the data. Source data are provided as a Source Data file.

conducted a crowd-sourced behavioral experiment in which participants (henceforth 'novices') were instructed to organize each of the 185 stimuli by the degree to which each image was representative of the hypothesized preferred category (face/body/scene) for each region (see Fig. 3a and Methods). In general, the responses made by novices were highly reliable (test-retest reliability > 0.8, tested on repeating a subset of 40 stimuli per participant, see Methods). We then measured the degree to which these category membership ratings correlate with the observed fMRI responses within the fROIs for each individual fMRI participant, and we compared this prediction to that of the model. A challenge in this analysis is that model prediction accuracies might be slightly overestimated given the distinct data used to lock down model parameters (from the opposite-hemisphere fROI) might not be fully independent because of correlated noise between homologous regions in opposite hemispheres. To avoid this potential problem, we tested the ability of models and humans to predict each fMRI participant's individual held-out data (instead of the pooled data averaged across participants, see section on Behavioral experiments under Methods). That is, for these analyses we used models not trained on any of the specific to-be-predicted subjects' data (or images), paralleling the situation of our human participants' predictions. Figure 3b shows that behavioral ratings do not predict the observed responses as well as the encoding models do (mean ± s.e.m across fROIs, 0.82 ± 0.01 for the ANN, 0.64 ± 0.04 for the behavioral data from novices, $P = 0.03$, Wilcoxon signed-rank test) indicating that the human intuitions of graded category membership do not explain the observed fMRI responses as well as the ANN-based models.

The previous analysis demonstrates that the models do not merely provide image-computable versions of the word-level descriptions of the responses of each region, but embody further

information not entailed in those word models. Here we ask whether the models know more about these regions than even experts in the field do. To find out, we invited professors who have published extensively on the human ventral visual pathway to predict the magnitude of response of the FFA, the PPA and the EBA to each of the 185 stimuli in our set. Strikingly, even the expert predictions were not as accurate as our (ANN) encoding models (Fig. 3b, mean ± s.e.m across fROIs, 0.82 ± 0.01 for the ANN models, 0.77 ± 0.01 for the experts, $P = 0.03$, Wilcoxon signed-rank test between between model and expert predictions across the fROIs). The difference between the ANNs and experts was even more striking when we removed the category-based variance by testing the predictions on individual images within each of the individual categories (Fig. 3c, mean ± s.e.m image-level predictions for individual images within categories across regions, 0.39 ± 0.03 for the ANN models, 0.23 ± 0.02 for the expert predictions, and 0.12 ± 0.03 for the novice participants; $P = 2.1 \times 10^{-5}$ between ANNs and experts, $P = 1.8 \times 10^{-5}$ between ANNs and novices, $P = 4.4 \times 10^{-5}$ between experts and novices, Wilcoxon sign-rank test). This was true even for the hypothesized preferred category for each region (Fig. 3c, mean ± s.e.m image-level predictions for the hypothesized preferred category across regions, 0.50 ± 0.04 for the ANN model, 0.17 ± 0.07 for the expert predictions, and 0.03 ± 0.10 for the novice participants $P = 0.03$ between ANNs and experts, $P = 0.03$ between ANNs and novices; $P = 0.03$ between experts and novices, Wilcoxon signed-rank test between model predictions, and expert and novice subjects across fROIs). Together these results demonstrate that current ANN-based models of the ventral stream provide more accurate predictions than the judgements of both novice participants and experts who have extensively investigated these regions.

The fact that our models make accurate predictions and operate directly on image pixels (i.e., are image computable) enables us to use these models to ask new questions. Next we use the models to put the claims of the hypothesized category selectivity of these regions to their strongest test yet.

**ANN models of the ventral stream enable strong tests of category selectivity.** Now that we are equipped with computationally precise encoding models that can predict the responses in the FFA, PPA, and EBA with high accuracy, can we connect these findings in neuroscience to other fields, which use verbal descriptions of category selectivity? To do this we first need to know if the previously claimed category selectivity of each region is even true, given that the responses in these regions have been tested for only a very small subset of possible images. Category selectivity can be defined and quantified in several different ways, but the most common definition concerns the category of images that evoke the highest response in a neuron or voxel or region. According to this criterion the selectivity of the FFA for faces could be falsified if any of the stimuli producing the highest responses in this region are not faces (as judged by humans).

So, are faces in fact the stimuli that produce the highest response in the FFA, places in the PPA, and bodies in the EBA? Here we cycle back to use the ANN-based models to put the category selectivity of these regions to a strong test by using models to simulate high-throughput experiments that could not be run with actual fMRI measurements. Specifically, we used our predictive models to screen ~3.5 million stimuli ($N = 3,450,194$ images, Fig. 4a) from three popular natural image databases to find the images predicted to produce the strongest response in each region. These datasets include VGGface, which consists of faces only, usually used to train models on face discrimination tasks; Imagenet, a diverse stimulus set with 1000 different object categories; and Places2, a stimulus set with snapshots of 400 unique place categories usually used to train models on scene categorization tasks. A histogram of the predicted responses of each fROI for all ~3.5 million images is presented for each stimulus database in Fig. 4b. The histogram for the FFA for instance, shows that the responses to (face) stimuli from the VGGface database is considerably higher than the response to stimuli in the other databases. The key question though is whether any of the top-predicted images based on this high-throughput screening procedure are not members of the hypothesized preferred category. To find out, we first visually inspected each of the top-predicted 5000 images for each region. Remarkably, we found that all were unambiguous members of the hypothesized preferred category for that region (five representative images are shown in Fig. 4c). To test farther down the list, we sub-subsampled the top two images from each thousand of the top 100,000 images for each region (Figs. S7-S9). Again, we found that all 200 such images were unambiguous members of each region's hypothesized preferred category.

Finally, for the case of faces and the FFA and PPA, we performed an additional test wherein we removed all the images in the VGG-face image set (that the builders of that set labeled as faces), and all the images in the Places2 image set (that its builders had previously labeled as places). Remarkably, when we asked each model to report the remaining 5000 most preferred images in those sets, the FFA model still ended up finding only stimuli containing faces and the PPA model still only found stimuli containing places (Figs. S10-S10). Had we found any stimuli predicted to produce a strong response in a region that were not members of the hypothesized preferred category for that region, we would have cycled back to scan participants viewing those images to see if indeed they produced the predicted high

responses. But we did not find such images, so there were no potentially hypothesis-falsifying stimuli to scan. This finding further strengthens the inference that these regions are indeed selective for faces and places.

Is this observation guaranteed, given that these regions were defined by a preference for one category over others? To investigate whether the alternative outcome was even possible, e.g., that the images predicted to produce the highest response in the FFA would not be faces, we performed a simulation of the experiment run on our human participants but instead on single units within a control ANN model. Briefly, we identified putative "face units" from the conv-5 layer of Alexnet based on snapshots from the dynamic localizer used in our experiment and built a ResNet-50 based computational model to predict the response to the 185 images averaged over these putative face units. In this case we find that despite choosing the model units (from Alexnet layer conv-5) that were putatively "face-selective" based on the higher response to faces than to bodies, scenes, and objects on the localizer task, 85% of the top-predicted images for those units are not faces (as compared to 0% for the human FFA). This simulation demonstrates that our method is capable of falsifying previously observed selectivities and that the model-derived "face units" were not as face-selective as the human FFA (Fig S12).

As we demonstrated above, the high-throughput screening strategy is a powerful way to test category selectivity, but it still depends on having the stimuli capable of falsifying the hypothesis in the screened stimulus databases. To address this limitation, we turned to a complementary image synthesis method. Specifically, given each encoding model (above), we used a generative adversarial network (GAN) to synthesize images that the model predicts would strongly activate each fROI (Fig. 4d). This method allows exploration of the naturalistic images space that is much broader than sifting through photographic databases (above). We found that the preferred images synthesized by the algorithm could be easily recognized as members of the previously hypothesized preferred category (Fig. 4e). Taken together, both the image screening and synthesis procedures demonstrate the power of the computational modeling approach that now enable us to strongly test and validate the claims of category selectivity for the FFA, PPA, and EBA on naturalistic images in a way that was not possible before.

**ANN models of the ventral stream enable efficient identification of features of the stimulus that drive neural responses in the fROIs.** Our previous analyses demonstrated how models of the ventral stream can be used to put theories of category selectivity to strong tests. But, as presented above, they still do not provide any human-interpretable intuition about which features of the preferred stimuli drive the responses within their respective fROIs. This is not only an important scientific question in its own right, but also one that could increase our confidence in the use of these models, by ascertaining whether the model is doing what we think it is doing (e.g., responding to face parts rather than incidental features associated with faces like hair or earrings). Distilling this human-interpretable intuition about each brain subregion directly from neuroimaging experiments has proven challenging because the standard approaches (reverse correlation or partial occlusion) require measuring responses to a very large number of images. Next we show how computational models may help overcome this critical barrier using a variant of Randomized Input Sampling for Explanation (RISE)[43]. Conceptually, this method relies on applying a large number of different occluding masks (2000 masks per image) to randomly subsample different parts of an image, obtaining the predicted response to each masked image for each fROI, and finally linearly combining the

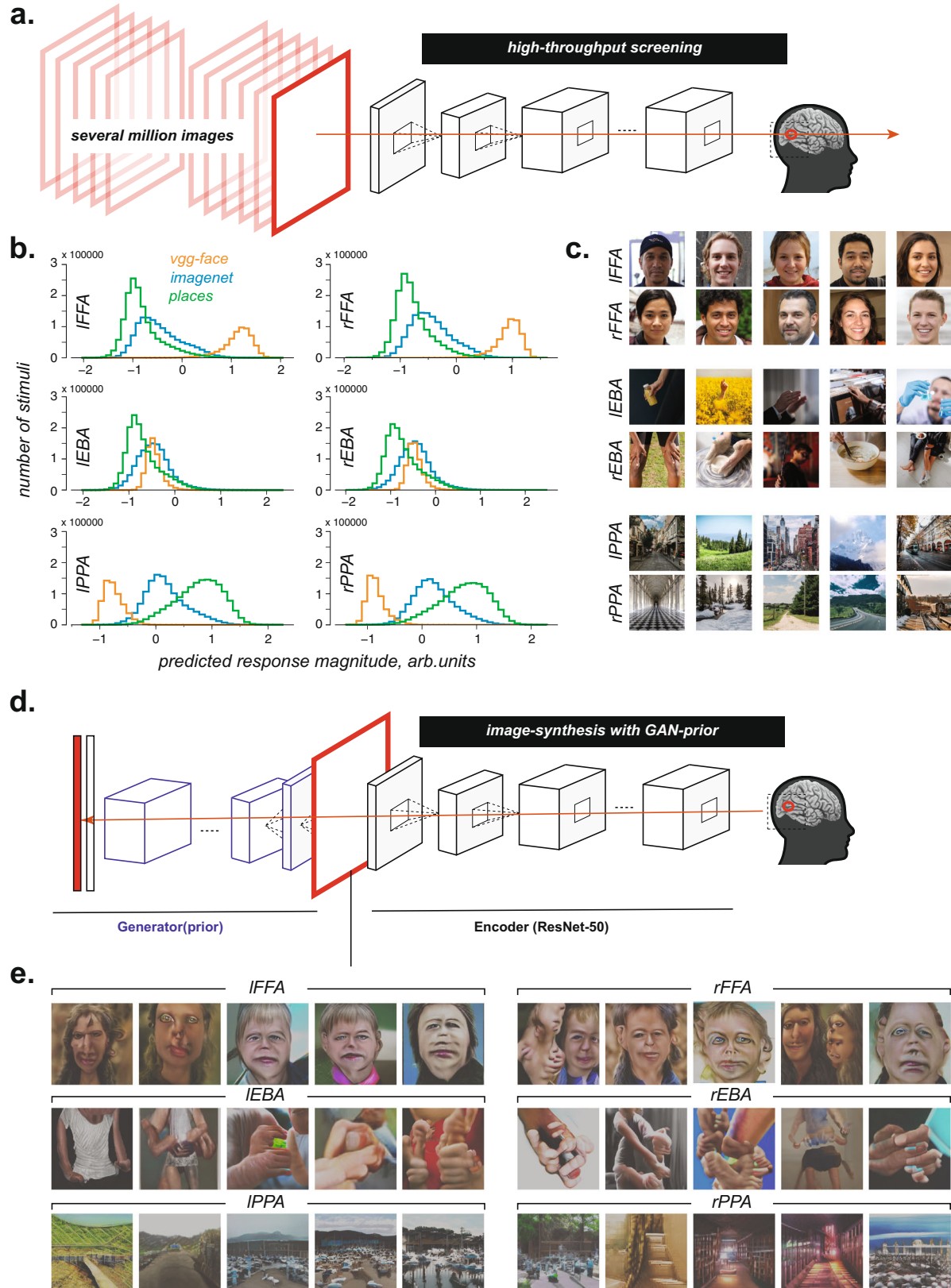

masked images and the predicted responses to generate an 'importance map' (Fig. 5a).

We first applied this method to identify regions of a face stimulus that trigger the response of voxels in the FFA. The overlaid colormaps in Fig. 5b suggest that the response of the FFA models are driven largely by eyes and noses. Similar analyses on the EBA (Fig. 5c) and PPA (Fig. 5d) suggest that the responses in the EBA models are driven by hands and torsos (and critically not faces), and the responses in the PPA models are driven by side walls and perspective cues. But our typical natural viewing

**Fig. 4 Strong tests of hypothesized selectivity using high-throughput screening and GAN-based image synthesis. a** Schematic of the high-throughput image screening procedure. Here we screened ~3.5 million stimuli from several public image databases through the encoding models and investigated the stimuli that the models predict strongly activate a given fROI. **b** Histograms showing the distribution of predicted responses by the encoding models for the different regions of interest. Each color indicates a different stimulus database for the predictions. Source data are provided as a Source Data file. **c** Representative images predicted to strongly activate each of the fROIs. See Figs. S7-S11 for stimuli subsampled from the top 100,000 images. Note that the actual images have been replaced with copyright-free illustrative versions here. See https://osf.io/5k4ds/ for the actual top images. **d** Schematic for the GAN-based image synthesis procedure. We coupled a Generative Adversarial Network (BigGAN) as the prior along with our Resnet50 (the encoder) to optimize pixels and synthesize new stimuli that predict yet stronger responses in each of the desired fROIs. **e** Stimuli synthesized from this GAN-based synthesis procedure that the models predict maximally activate the FFA, EBA, and the PPA.

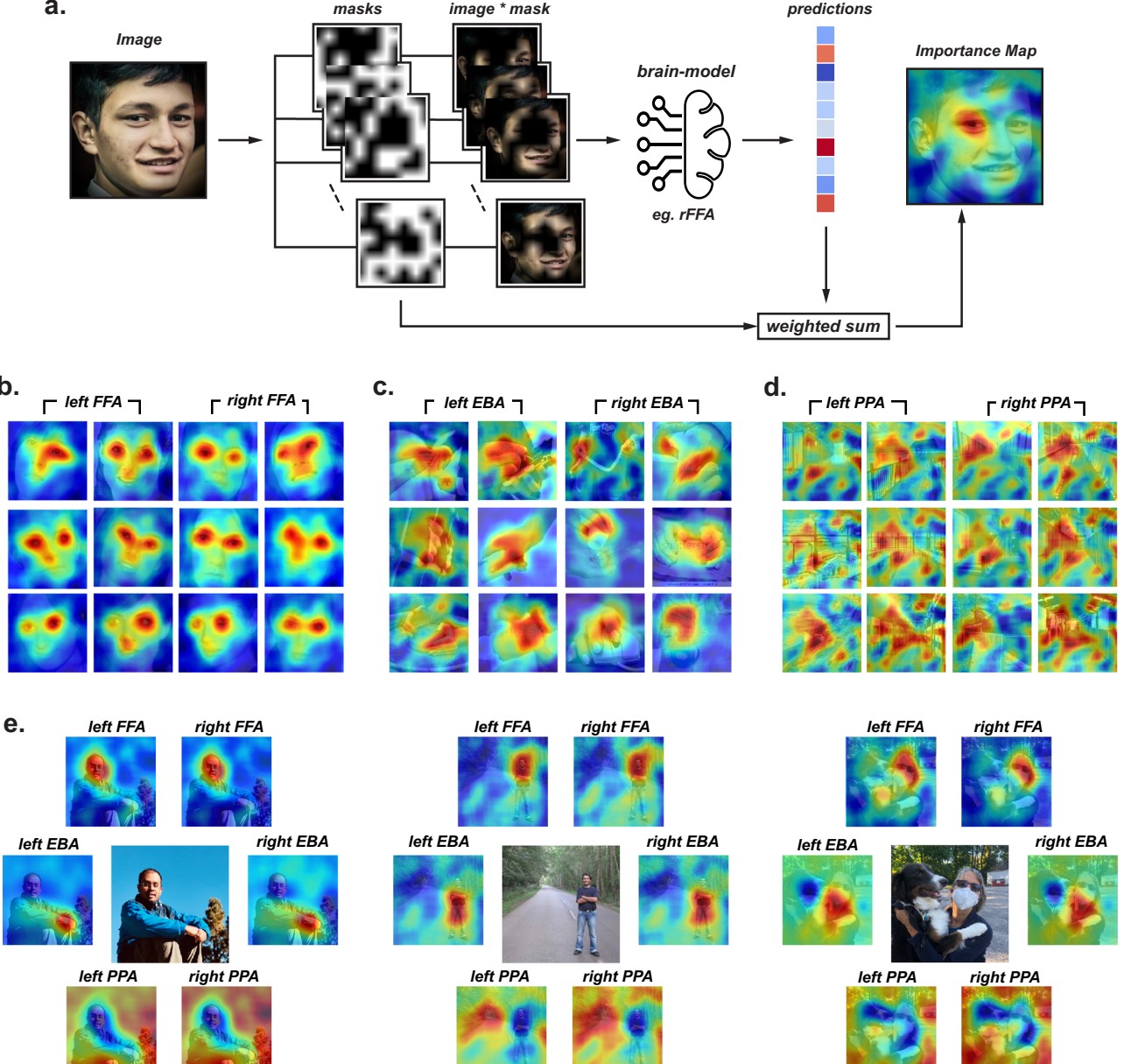

**Fig. 5 Using computational models to identify features of the stimulus that drive neural responses in each fROI. a** Schematic of the algorithm used to generate importance maps. We convolve each image with 2000 masks (left) and run the masked images through our brain model for each fROI to generate the predicted response for each image. The importance map (right) is the generated by taking a weighted sum of the masks with these predicted responses. Permission was obtained from the person in the photograph to publish the image used in this figure. **b** Importance maps for the left and right FFA for the images predicted to generate high responses. The warm colors on the overlaid colormaps indicate regions that most strongly activate the FFA and cool colors indicate regions that do not activate the FFA. **c** Same as **b** but for left and right EBA. **d** Same as **b** but for left and right PPA. **e** Importance maps for three representative photographs with faces, bodies and background scenes and importance maps for each fROI. Permission was obtained from the persons in the photographs to publish the image used in this figure.

experience rarely contains isolated stimuli like the maximally activating images used for these analyses. Do models for the FFA, EBA, and the PPA isolate the faces, bodies, and scenes from such composite images? We tested this question on complex natural images (Fig. 5e) and indeed as hypothesized, we find that computational models of the fROIs identify the expected region of the scene containing preferred stimuli for that region. Together these analyses demonstrate the utility of computational models in building hypotheses about which features of a stimulus are driving the predicted responses in a given region of the brain. Of course once any new hypotheses are derived from the models, they should be tested empirically with actual brain measurements.

## Discussion

In this study we built and tested computational models of three of the most extensively studied regions of the human ventral visual pathway, the FFA, the EBA, and the PPA. We find that particular ANN-based models of the primate ventral stream can predict the response of these regions to novel images with high accuracy. We also show that the predictions made by these models generalize across participants and are more accurate than prior descriptive models and even experts in the field. Most importantly, these predictive models enabled us to subject the claimed category selectivity of the FFA, PPA, and EBA to their strongest tests on naturalistic images to date. We used the models to screen millions of images, and to synthesize novel images, to ask whether any of the images predicted to produce the highest responses in each region are not members of the hypothesized preferred category for that region. In both cases, we failed to find such images, even though our approach has the power to do so, thus substantially strengthening the previous claims of face, scene, and body selectivity of the FFA, PPA, and EBA.

We are not the first to use ANN-based predictive models of neural responses measured with either neurophysiology[21,44] or fMRI[22,45–48]. What is new in our study is the high accuracy of our model predictions, their application to the mean response of entire cortical subregions (as well as patterns of response within those regions), their ability to generalize across participants, and most importantly, the ability to test longstanding hypotheses about the selectivity of each region. Our ANN-based models also provide efficient and high-throughput methods for discovering features within each preferred image that might drive each region. Our ANN-based models not only confirm, but also go beyond classic "word models" (i.e. that the FFA, PPA, and EBA are selective for faces, places, and bodies respectively) in two important respects. First, they are image computable, so we no longer need a human to tell us if an image is a face or place or body to predict if a brain region is likely to respond to that image. Second, they provide fine-grained predictions of the responses to individual images both within and outside the preferred category. As we show, these models outperform both binary and graded versions of the category selectivity hypothesis, and thus "know" more about the selectivity of each region than is entailed in the simple category-selective word label. Further, the models' ability to outperform even scientific experts who have extensive experience measuring the responses of these regions shows that these models contain new information about these regions not already built into the intuitions of experts in the field. We provide the vision community with this new synthesized knowledge as downloadable models of entire cortical regions, as well as with a roadmap for testing their own hypotheses of the FFA, PPA, and EBA.

Given all of these advantages of the ANN-based encoding models, should we now dispense entirely with word models? We think not. Because word-based hypotheses are grounded in everyday concepts, they provide accessible and intuitive explanations of the function of each cortical region. Further, theories in psychology and development and evolution concern real entities in the world like faces and places and bodies, and their significance to humans, which cannot be fully captured by vectors of connection weights. Word-based models are therefore necessary to interface between the rich tradition of findings and theories from those fields, and the empirical findings in neuroscience. For all these reasons, ANN-based computational models complement rather than replace word models. Minimally, the words "face", "body", and "place" serve as pointers to the code that can be used to execute the current models of each brain subregion.

Although ResNet50-V1 provided the numerically most accurate models across regions (consistent with ref. [38]) based on a broad screen of models it is important to note that a single study or small number of regions considered (as in our own work) is insufficient to prescribe a single base-model as being the most brain-like. Ultimately the model arbitration will require a community-wide effort and rigorous integrative benchmarking on completely independent data from new subjects and evermore regions (similar to say the BrainScore platform[21,33] for non-human primate data). An important contribution of our work is the fROI-scale of computational modeling, which makes it possible to evaluate our exact model on completely independent subjects, hypotheses, and data. fROIs like the FFA, PPA, and EBA can be isolated in almost all participants and our models make testable predictions and are more directly falsifiable than say voxel-wise models (though as we show, these metrics are highly correlated).

That said, our current modeling efforts also show that there is room for improvement for example in the within-category predictions. One route will make use of new models now being developed that include known properties of biological networks[49–52] and that may better fit neural responses. Another route will learn from model failures in an effort to improve predictive accuracy. For example, the ANNs used here were trained on naturalistic images, are vulnerable to targeted adversarial attacks[53,54] and are known to have limited generalizations to out-of-domain sample distributions (for instance, do not generalize from natural images to line drawings). These shortcomings suggest that these models are not likely to accurately predict observed fMRI responses to more abstract[3,55,56] and symbolic stimuli[57–60] including contextually defined faces[61,62]. Thus an important avenue of future work will entail exposing and expanding the bounds within which these models retain their predictive power. An effective strategy could be to run evermore targeted experiments that push model-development efforts either by including more diverse set of stimuli for training (like sketches, line drawings, cartoons etc.), or stronger inductive biases as biological networks.

Further, our ultimate our goal as vision researchers is to build computational models that not only predict neural responses in a given region of the brain, but that also reveal the full sequence of computations over all successive stages of visual processing that lead to those responses. For example, future work can also test whether cortical regions thought to provide input to the FFA and PPA (e.g., the OFA and OPA, respectively) match earlier layers in the network. An example of this "simultaneous compare at all processing levels" approach has recently been taken up by the Brain-Score open model evaluation platform[33]. Thus, our current work is just the first iteration of a cycle of high-quality data collection and model improvement that should evermore accurately capture the computations entailed in visual information processing.

In this paper we have focused on building models that can accurately predict the mean response of each cortical subregion (i.e., as if it were a single "unit"). Of course, information is encoded in each subregion not by the mean response of millions of neurons, but by the pattern of response across those neurons.

Indeed, our models were also quite accurate in predicting responses of individual voxels and of patterns of responses across voxels (Figs. S5, S6), and prior work in rodents and macaques has shown that similar models have good predictive accuracy for even finer-grained recordings[25–27]. So we now have the methods in place to derive excellent encoding models for the entire visual pathway, at any spatial grain our data provide, and all transformations of those data (e.g., RDMs). This success represents substantial progress and opens the door to future work on the even bigger challenge of understanding how other parts of the brain read these neural codes to tell us what we are looking at. Because neural responses typically contain information not only about the stimuli that elicit the strongest response[63] (cf the nonpreferred predictivity found here), causal interventions on the brain are necessary to determine which of the information present in each region is read out to produce our rich repertoire of visually guided behaviors[64].

Perhaps most fundamentally, a rich scientific account of visual processing in the brain requires not only accurate predictive models, but an understanding of how those models work. New methods for analyzing computational networks, much like those developed by neuroscientists for understanding brains, are enabling us to demystify how ANNs produce the responses they do. We can characterize and visualize responses in different network layers[18,65] and "lesion" parts of the system to test their causal role in explaining neural data[66]. But with networks we also have a suite of new methods that are not available in human cognitive neuroscience: We can independently alter properties of the network architecture, loss functions, and training diet to test which of these properties are essential for the model to perform as it does[49,67]. Taken together, the highly accurate prediction of neural responses that are already possible, along with powerful methods for improving and analyzing our models, are giving us an unprecedented opportunity to discover and understand the actual neural computations that enable us to see.

## Methods

All studies were approved by the Committee on the Use of Humans as Experimental subjects of the Massachusetts Institute of Technology (MIT).

**Functional MRI**. All the fMRI scanning sessions were performed at the Athinoula A. Martinos Imaging Center at MIT on a Siemens 3-T MAGNETOM Prisma Scanner with a 32-channel head coil. Functional scans included T2*-weighted echoplanar (EPI) BOLD images for each experiment session (acquisition parameters: simultaneous interleaved multi-slice acquisition (SMS) 2, TR = 2000ms, TE = 30 ms, voxel size 2 mm isotrotropic, number of slices = 52, flip angle: 90°, echo-spacing 0.54 ms, 7/8 phase partial Fourier acquisition whole brain coverage). Each fMRI participant ($N = 4$, 2 females) participated in five neuroimaging sessions. A high-resolution T1-weighted (multi-echo MPRAGE) anatomical scan (acquisition parameters: 176 slices, voxel size: $1 \times 1 \times 1$ mm, repetition time (TR) = 2500 ms, echo time (TE) = 2.9 ms, flip angle = 8°) was acquired during the first scanning session, along with data to localize the functional regions of interest (fROIs). The data to be modeled were collected over 4 additional neuroimaging sessions (sessions 2–5) using an event-related experiment paradigm. Details of the (dynamic) localizer and event-related scans are described next.

**Dynamic localizer**. The fusiform face area (FFA), parahippocampal place area (PPA) and the extrastriate body area (EBA) were localized in each individual participant using a standard dynamic localizer, which has been used extensively to isolate these regions[28,30]. Briefly, the stimuli were short video clips corresponding to one of five stimulus classes (faces, bodies, scenes, objects and scrambled objects). Each experiment run included 25 18-s blocks (20 stimulus blocks, four per category, and 5 fixation blocks). Each block contained six 3-s long video clips randomly drawn from a set of 60 clips. The stimuli were 20 degrees of visual angle (dva) wide and 15 dva tall. The order of the conditions was palindromic (e.g., A-B-C-D-E-E-D-C-B-A) and subjects were given no specific instructions except to view the videos being presented on the screen. Each subject participated in five runs of the dynamic localizer experiment over the course of the first neuroimaging session.

**Event-related experiment**. We used an event-related experimental paradigm for the main experiment. The stimuli comprised naturalistic full-field, unsegmented,

colored images sampled predominantly from the THINGS database[68]. The stimulus set included 25 images with faces, 50 images with bodies, 50 images with scenes, and 65 images with objects (total $N = 185$ images). Each image subtended 8dva inside the fMRI scanner and participants viewed 100 unique images per experimental session. Each stimulus was presented for 300 ms followed by minimum inter-stimulus-interval (ISI) of 3700 ms and maximum ISI of 11700 ms optimized using OptSeq2 (https://surfer.nmr.mgh.harvard.edu/optseq/) and the stimulus order was randomized separately for each experiment run. All participants provided informed consent before being scanned. Subjects were instructed to maintain fixation on a small 0.3dva fixation cross at the center of the screen at all times, with no other task. Subjects viewed a fixed set of 100 unique images, 15 of which were repeated in every experiment session (normalizer images) and used to normalize the data (see data normalization below). Each stimulus was repeated at least 20 times per participant to maximize the test-retest reliability of the data (see Fig S1). Subjects performed 10 experimental runs per session and the data were collected over four neuroimaging sessions.

**fMRI preprocessing and general linear modeling, and data normalization across sessions**. fMRI data preprocessing was performed on Freesurfer (version: 6.0.0; Downloaded from: https://surfer.nmr.mgh.harvard.edu/fswiki/). Data preprocessing included slice time correction, motion correction of each functional run, alignment to each subject's anatomical data, and smoothing using a 5 mm FWHM Gaussian kernel. Generalized linear modelling (GLM) for the dynamic localizer was also performed on Freesurfer and included one regressor per stimulus condition, as well as nuisance regressors for linear drift removal and motion correction per run, and analyzed on the surface reconstructed versions of the data. GLM analysis for the event-related experiment was performed using GLMdenoise[69]. This method, optimized for event-related fMRI, estimates the noise regressors directly from the data. Consistent with previous reports[69,70], this method substantially improved the test-reliability of the estimated beta parameters in our pilot experiments. Using this method, we estimated a single beta parameter estimate per stimulus corresponding to the change in BOLD signal in response to the presentation of that image. Following methods routinely used in non-human primate studies[26,71] we normalized the data per group of 100 images by the responses observed to the 15 normalizer images repeated per experiment session (by subtracting the mean and dividing by the standard deviation).

**Encoding models**. We screened several computational models, predominantly deep convolutional artificial neural network (ANN) based models to predict the observed responses (normalized beta estimates per image, see above) in the FFA, PPA and the EBA to the images in our stimulus set. Specifically, we used ANNs trained to categorize static images based on a large dataset of labeled natural images. Our working hypothesis is that these networks learn internal representations similar to those produced by the population of neurons in the brain. Given that the learned representational space in these computational models may not exactly align with those in the brain, we allow a single linear transformation on those features to create a mapping between the ANN model and the neural measurements[20,39,44]. Therefore, our predictive modeling methods comprised two parts: (a) an embedding model (e.g., a given ANN model) and (b) a linear mapping function, each of which is described next. Note that all the predictions reported in the paper are always based on images not used to train the model (cross-validated) and with the specific layer choice and hyperparameters determined using completely distinct neural data from the homologous regions in the other hemisphere (see outline in Fig S13).

*Embedding model*. ANN-based models trained to categorize stimuli from static natural images have previously been shown to develop representations that are remarkably similar to recordings from monkey and human visual cortex[19,20,22,26,44,45,72]. In our experiments we screened a set of 60 different computational models at predicting measurements (Table S1) from the human visual cortex. Parameters of each network (except the randomly initialized untrained Resnet-50 model) were first optimized for object categorization performance using supervised learning methods[15]. At the end of this training, all the parameters were fixed and the internal activations at the individual layers of the network were extracted to predict the neural responses.

*Mapping ANNs to measured responses*. As detailed above, we used linear mapping methods to make predictive models of neurons using features extracted from different layers in each embedding model. We used two linear mapping methods to predict neural measurements from ANN activations. Each of the mapping methods are explained below.

## Choice of base-model and best model layer

Each neural network encoding model (base-model or network) has several layers of computations. Activations at each layer constitute a different feature set that could be related to the neural responses in each of the ROIs. We first screened all the feature sets (every layer of every base-model). To do this, we used a regularized ridge regression with 5-fold cross-validation (over five separate randomization seeds) to map the activations from each layer of every network to the neural responses in each fROI. We used this

method because of its low computational cost that allowed us to evaluate and compare many feature sets across many networks (Figs. S3, S5). We did not perform any additional search for the optimal regularization parameter for the ridge regression and set it to 0.01 (further post-hoc analyses show that changing this parameter within reasonable limits does not change the relative ordering between models). These screening prediction scores were used to freeze the decisions about the specific base-model and the specific layer before obtaining the final predictions for all subsequent experiments.

The choice of the base-model was decided through an integrative benchmarking approach[33]—that is, we chose the base-model that simply had the numerically highest cross-validated predictive accuracy (always based on held-out data) across all the 6 fROIs considered in the study. Note that this choice is highly constrained and does not change even when N-1 fROIs were used to make the selection for the Nth fROI.

For that specific base-model, the best layer for a given fROI (for example, the left FFA) was determined based on distinct neural data from the homologous region in the other hemisphere (the right FFA in this example). This is how we ensure that all the free parameter choices (base-model architecture and specific layer) for a given region were fixed based on distinct neural data before determining the final cross-validated prediction accuracies. Moreover, the goal of this exercise was to not arbitrate between models (see Discussion) but to identify a sufficiently good representational feature basis set for subsequent experiments.

## Fitting the fROI responses from activations in the best layer

After selecting the numerically highest performing base-model and model layer based on distinct neural data (see above) we fit a two-stage linear mapping function[26,73] to predict the brain measurements from the best network and layer. In this method, the mapping function consists of two separate stages that essentially codes the what and where of the neural function. The layer activations for an individual image consist of three dimensions, width, height, and depth. The width and height dimensions correspond to the spatial dimensions of the input while the depth dimension refers to the various encoding dimensions (i.e., convolutional filters) in that specific layer. The first stage of mapping, which learns the spatial correspondence between the model and the brain activity, consists of learning a unique spatial mask over the width and height dimensions of the activation map that best relates the model activations to the fROI responses. In the second stage of mapping, the spatially masked activations are then averaged over the spatial dimensions (i.e., the width and height of the activation map) and then multiplied by a second vector of learnable parameters that computes a weighted average of the different encoding dimensions (i.e., convolutional filters). The mapping function parameters are optimized to reduce the prediction error for the brain responses within each fROI. Because this method allows for per-fROI spatial masking of the feature set, it contains much fewer trainable parameters (compared to most regression methods such as ridge regression) and leads to more accurate predictions. However, this comes at a relatively high computational cost compared to the linear regression and was therefore only computed once we determined the best base-model and layer after the screening procedure.

The convolutional mapping has two hyperparameters that control the degree of weight regularization for the spatial and encoding dimensions. We considered 6 points, spaced evenly on a logarithmic scale, within the [0.01–100] range. We chose these hyperparameters by performing a grid-search on these two hyperparameters using a separate 10-fold cross-validation procedure. As before, the grid-search was performed on distinct neural data from the homologous region in the opposite hemisphere (on the same set of 185 images) and on the model layer determined from 1) above. The final model prediction accuracy for the fROI was then determined (using features from the specific model layer, and using grid-search regularization weights determined from homologous regions from the opposite hemisphere) using a single 10-fold cross-validation (using a different random seed for both weight initialization and data split from before) over the 185 images. This entire procedure is outlined schematically in Fig. S13a.

Note that we always used distinct neural data to determine all the free parameters (choices on which base-model, layer and grid-search hyperparameters) to reduce any model selection bias. Nonetheless it may be argued that neural data across hemispheres and regions are not entirely independent because of perhaps shared scanner noise even across hemispheres within each individual fMRI participants' data (see ref. [74] for instance). While we cannot entirely rule out the influence of this shared noise on the reported neural predictivity estimates, these effects, if any, are likely small and do not qualitatively affect the majority of results. For quantitative comparison of model prediction accuracy to humans, we avoid this problem by cross-validating across both images and participants (see section on Behavioral experiments below).

*Noise-ceiling estimates.* The noise-ceiling indicates the maximum attainable accuracy of the encoding models given the reliability of the recorded data themselves. To derive these estimates, we randomly split the 20 runs of recorded data into two groups (and repeated this randomization 10 times). The noise-ceiling for every subject was estimated by first computing the beta estimates per image separately for each half of the data, and then measuring the (Spearman-Brown corrected) correlation between the estimated betas from the two halves (repeating this entire procedure 10 times for each binary split). For the pooled data we used the same procedure to measure the pooled beta-parameters (by averaging across the per-image beta estimates in each split over subjects) for each randomization split and calculating the Spearman-Brown corrected split-half correlation between the estimated betas across splits. The plots in Figs. 2, 3 indicate the mean and the standard

deviation of the correlations across splits. For the across-subject prediction tests we estimated the noise-ceiling directly by correlating the pooled data from (n-1) subjects with the data for the nth subject (Fig. S4, across-subject generalization test 1) and by correlating the observed data for each subject with the observed data for every other subject (Fig. S4, across-subject generalization test 2).

**Neural predictivity**. We used the Pearson product moment correlation between the observed responses across images and the cross-validated predictions (predictions on held-out data) of responses to those same images (neural predictivity score) to assess the match between the model and the brain. Given the question we were addressing, we used the models to predict the activations of each individual voxel (voxel-wise predictivity), the mean response within a given fROI for each subject (mean-fROI predictivity), or the pooled response in a given fROI across subjects (pooled predictivity).

**Population code representational similarity analysis (RSA)**. We used representational similarity analyses to compare the degree to which the voxel-wise pattern of population responses observed in the brain within the fROIs matched the voxel-wise pattern of population responses predicted by the encoding models. For every (observed or predicted) nvoxels × nstimuli data matrix corresponding to a given fROI, we computed the Euclidean distance between each pair of images ($^{185}C_2 = 17020$ pairs) to obtain the observed and predicted $185 \times 185$ images representational dissimilarity matrices[75–78] (see ref. [79] for a discussion on why Euclidean distance is better suited than the oft used correlation distance for responses in the FFA, PPA, and the EBA). The similarity between the observed and predicted RDMs was assessed by taking a spearman correlation between observed and predicted RDMs.

**Behavioral experiments**. To address the degree to which the observed responses could be predicted based on human intuitions of graded category membership, we conducted a behavioral experiment on an independent group of participants recruited from Amazon Mechanical Turk. Three independent groups of participants performed each of the face/body/scene tasks. Participants in the experiment were directed to an online platform (www.meadows-research.com) where they performed a drag-rate arrangement task. Specifically, the subjects were instructed to "Drag and drop the images based on how [face/body/scene]-like they are" by organizing each of the 185 images onto a graphical region where the top of the y-axis indicated very [face/body/scene]-like images and the very bottom of the y-axis indicated that the images were not [face/body/scene]-like at all. To make sure that participants comprehended the instructions and were indeed performing the task consistently, they were asked to repeat the experiment on a smaller subset of 40 images. Participants took on average 15 minutes to finish the experiment and were monetarily compensated for their time. The participant responses were in general reliable but we included only participants with a mean test-retest correlation 0.85 (Total number of subjects recruited per task: 130, Total number of participants' data retained, N = 106 subjects on the face-task, N = 115 on the body task, N = 60 for the scene task, mean test-retest correlation 0.95, 0.95, and 0.90 for the face, body, and scene tasks respectively). The match to observed data was assessed by taking the Pearson correlation between each novice's ratings and the observed fMRI responses in each fMRI participant's individual fROIs (then averaged over fMRI participants to get the accuracy of that novice).

To assess the degree to which experts could predict the observed responses to the 185 images, we reached out to a group of 30 researchers who we judged to have the greatest expertise worldwide in measuring fMRI responses from the human ventral visual pathway. Ten of them provided responses (all were Assistant Professors or above). These experts were asked to participate in a similar drag-rate arrangement experiment as described above, but with different instructions. Experts were instructed to "Place the images on the graph based on the expected [FFA/EBA/PPA] activations by organizing each of the 185 images onto a graphical region where the top of the y-axis indicated maximum [FFA/EBA/PPA] activation and the very bottom of the y-axis indicated low [FFA/EBA/PPA] activation (Fixation)." Like the novice subjects, the experts also repeated the task for small subset of 40 images. The order of the tasks was randomized across experts who took on average 30 min to finish the experiment. Responses were highly reliable within experts (mean test-retest reliability 0.97, 0.96, and 0.96 for the FFA, EBA, and PPA tasks). As with the novices, the match to observed data was assessed by taking the Pearson correlation of each experts' own predictions, with the observed response in each fROI for every fMRI subject.

In order to fully allay any concern about prediction accuracies being biased in favor of models (see last paragraph on Encoding Models above), we used models cross-validated across both participants and images (similar to Fig. S4, third column from left). To expand, for each subject, we used computational models trained using a 10-fold cross-validation on all other subjects. This procedure is outlined schematically in Fig. S13b. In this way the models did not use any of that Nth fMRI participants' individual data (nor those specific images on which the predictions are made) in the model training procedure. For models, the match to observed data was assessed by taking the Pearson correlation between each (Nth) subject's individual data for every fROI, and the predicted responses by the model trained on N-1 subjects (and then averaged over all fMRI subjects).

**Image screening**. Given that our trained encoding models for each fROI could operate directly on the images (i.e., were image computable) and could predict the observed responses within the fROIs for naturalistic stimuli at relatively low computational costs, we screened several large publicly available image databases to find stimuli that the models predict would strongly drive responses in a given functional region of interest. Specifically, we screen the images from the IMAGENET[35] database which contains a diverse set of images from 1000 stimulus categories based on WordNet ($N = 1,281,167$ images), the Places2 database[36] which contains stimuli from about 400 unique scene categories ($N = 1,569,127$ images), and the VGGface2 database used to train models on face recognition with several stimuli containing faces only ($N = 599,900$ images, Total images across all 3 databases = 3,450,194). In all cases we ran all ~3 million stimuli through the models and obtained the model-derived predictions and visualized the subsampled top 100,000 stimuli predicted to activate a given fROI.

**Image synthesis**. We used the predictive model of the brain responses to synthesize natural-looking stimuli that were predicted to elicit large responses in each fROI. We used a synthesis method based on[80] which formalizes the image synthesis procedure as an optimization problem. Our goal was to synthesize stimuli that would maximally activate a given target fROI. For this, the optimization objective is defined as the magnitude of the predicted response in the target fROI. Compared to prior work[26] where the pixel values were iteratively tuned following the ascending gradient of the optimization objective, here we instead used an alternative parameterization of the image based on Generative Adversarial Networks (GANs)[81]. This reparameterization is done via a generative neural network [specifically, BigGan[82]] that is trained to produce natural-looking images given a low dimensional latent code and a 1000-way image category. The complete network used for image synthesis consisted of the generative network followed by the encoding neural network and the linear mapping function (see Fig. 4). The input to this network consisted of the GAN latent code and a 1000-way class identity and the output was the predicted brain activity in the target fROI. We used the pretrained version of BigGan network, which was trained to produce images of size $256 \times 256$ pixels (available via Tensorflow hub). The class category variable was initialized as $\alpha \mathcal{S}(n)$ where $\alpha = 0.05$, $\mathcal{S}$ is the softmax function, and $n$ is a vector of scalars randomly sampled from the truncated normal distribution between [0, 1]. The latent vector $z$ was also sampled from a truncated normal distribution between $[-2, 2]$. The *truncation* parameter was set to 0.5. For each synthetic image, we iteratively minimized the objective function for 30000 steps using the Adam optimizer[83] with a fixed learning rate of 0.001[34,36].

## Data availability

Stimuli, model checkpoints, preprocessed fMRI data, and behavioral ratings associated with experts and novices are available at the Open Science Framework repository for this project (https://osf.io/5k4ds/). The following publicly available resources were used for this work: THINGS database: https://osf.io/jaetd/ Imagenet: https://image-net.org/download.php Places2: http://data.csail.mit.edu/places/places365/train_large_places365standard.tar VGGFace2: http://www.robots.ox.ac.uk/~vgg/data/ Source data are provided with this paper.

## Code availability

The code used to generate the plots, model checkpoints, and model weights are available at https://osf.io/5k4ds/. Code to run inference on custom images directly from the browser will be made available from the authors' Github page (https://github.com/ratanmurty/NatComm2021).

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

## Acknowledgements

We would like to thank Martin Hebart for early access to the THINGS Database and members of the Kanwisher and DiCarlo labs for feedback on the project and drafts of the manuscript. This work was supported by the NIH Shared Instrumentation Grant S10OD021569 to the Athinoula Martinos Center at MIT, the NSERC Discovery Grant RGPIN-2021-03035 and the HBHL startup supplement grant HBHL-2b-NISU-13 (to P.B.), the MIT UROP Direct funding and the MIT Quest for Intelligence undergraduate funding (to A.A.), the Office of Naval Research grant MURI-114407 and the Simons Foundation Grant SCGB-542965 (to J.J.D.), and the NIH Pioneer Award NIH DP1HD091957 and the NSF Science and Technology Center—Center for Brains, Minds, and Machines Grant NSF CCF-1231216 (to N.K.).

## Author contributions

N.A.R.M., P.B., J.J.D., and N.K. conceived and designed the study. N.A.R.M. and P.B. collected human fMRI data, N.A.R.M. and A.A. collected and analyzed the human behavioral data. N.A.R.M., P.B., and A.A. analyzed the data and built the computational models. J.J.D. and N.K. supervised the project, N.A.R.M., P.B., A.A., J.J.D., and N.K. wrote the manuscript.

## Competing interests

The authors declare no competing interests.
