## [Peer Review File · Nature Communications]

REVIEWER COMMENTS

Reviewer #1 (Remarks to the Author):

The paper, "Computational models of category-selective brain regions enable high-throughput tests of selectivity", is an easy to read, clearly argued paper in which the authors present a convincing case that FFA, PPA and EBA are indeed selective for faces, places and body parts.

They do this by generating encoding models that allow for the image-computability of faces, places and bodies using DNNs and BOLD-MRI data. These models have an impressive performance in their capability of predicting out sample (both stimuli and subject).

Using image-computability models they show that no black swans are found (e.g. no exemplars of the wrong category) that would challenge the idea of category selectivity for places, scenes and bodies.

Furthermore, images generated using these models are hyper-exemplars of their categories.

The sum of these results make this a very convincing paper for the, in itself unsurprising, main point (indeed exemplar specificity). However, in doing so they authors present an approach to determine the selectivity of brain areas in a computational manner, opening up an avenue of possibilities.

However, I do have some (somewhat) more major, and minor remarks.

Somewhat more major remarks

L200-L203, it is not clear to me if, for the voxel models, the mean ROI response (for a stimulus) is subtracted from the voxel responses. This would be necessary to make a claim on the voxel information above, and beyond, the ROI. That a voxel contains some information from the ROI is a statistical necessity that is, in itself, uninformative.

L218-L236, this is a fun experiment, but I am wondering if this is not very much begging the question. What does it show. The authors suggest it might indicate (I guess if novices performance match the model?) a graded version of category selectivity but I miss the necessity of this reasoning.

L238-L261, the fun experiment continues, but basically the same question, what would it mean if experts were better, or equal, at predicting the BOLD-response to these categories than a BOLD-predicting model. For both the novel and expert I get that it showcases the superior performance of the model, but does it anything beyond this?

Minor remarks

L125-L130, the vast majority of networks tested are feedforward networks, with resnet-50 being a strong 'winner'. Given the little difference between the top performing model this is not a big point but there is an emphasis on the CorNet's in these lines that are not justified by the tests performed, or the results discussed.

L301-L303, how far do you need to go in the list before the first black-swan appears (as a percentage of

all images tested), and what happens beyond this, a gradual transition of face-like, or a more abrupt transition (and fair enough, this is difficult to characterise)?

L492-L494, all subjects have robust L and R EBA, FFA and OBA's. Where they selected for this, and/or is this guaranteed to happen with enough localisers?

Fig 2. The title is models , but only the resnet-50 results are shown. I believe the rest of the top-10 networks would be very similar but these are not shown. Is it not better to just refer to it as resnet-imagenet-50 (or an abbreviation of this)?

I really enjoyed this paper, best regards, Steven Scholte

Reviewer #2 (Remarks to the Author):

Computational models of category-selective brain regions enable high- throughput tests of selectivity

This paper measured the response properties of category-selective regions with a new high quality fMRI dataset, and then accurately modeled these responses using a deep convolutional neural network encoding-model approach. These image-computable models accounted for overall response profiles better than current intuitive models of what these regions do (with several other thoughtful tests into the different facets of responses of these regions). These models then allowed the authors to conduct a high-throughput test of the category-selective process, finding no rogue non-category selective images in a set of 3.5M would likely yield a high response in these regions. And, they probed the nature of the features encoded in these regions leveraging some of the newest advanced methods available.

The paper is clearly written; the scope of the question at hand is nicely targeted, while the the methods applied to this question are broad, ranging from measures of everyday intuitions to sophisticated, cutting-edge techniques like GANs for image synthesis; the analyses are sound; the patterns of results are strong; and the claims are clear, important, and well-justified.

I also think there are a rather substantial number of patterns of data that could be further scrutinized (!) -- far beyond a single paper's worth of ideas here. As such, I am not going to tug on the theoretical threads of these datasets that I think could use more exploration (likely ideosyncratic to me), and instead will focus solely on the claims directly presented. To this end, I have only minor comments. (Relatedly, it is great that the authors are providing the data to the community for further exploration.)

How good are these models? My main comment is that I was left by the end without a clear sense of the answer to whether these encoding models have completely nailed the responses, and to what degree there is representational room to grow, as measured in this fMRI dataset. For example, figure 2 a-c are rather outstanding—I think, this is pretty close to perfect. But, figure 2f it seems like there may be pretty far to go for some categories in some regions. Especially, when in figure 3c, the same data (I think?) is presented, but this time with the noise ceilings so it's easier to tell what's measurement noise vs uncaptured structure. Is there important information to be gained by the fact that e.g. the within-face

predictivity is much higher than the within-body predictivity in FFA, for example?). I think the main take-home about the success of the encoding models for these regions could perhaps be clarified with some sort of synthesized summary of what they get right (which is a lot, obviously) but maybe also what your current analyses indicate that they are not capturing, given the reliable structure of responses to these 185 images.

Generalizing from one subject to another subject. When reading these results I wanted a little more context: how well is the encoding model transferring from one brain to another, given how well that brain directly correlates to the other brain? For example, is it at the noise ceiling of the data, or is there reliable structure measured in the FFA between subjects that the encoding model isn't capturing? Right now I cannot tell apart how much these analyses are telling me about the similarity and differences of people's FFAs, from the effectiveness of the encoding model itself.

Model comparison. One of the first steps was the model screening procedure, with a large number of models compared. There are some interesting models in there like "curvature" and "room-layout" that don't seem like deep net models (but maybe they are?). In any case, I wish there was a little more information/meta data about each of these models, accompanying Fig S5).

Methods Clarifications

Can you clarify how the encoding models were fit for these finer-grained analyses for within-category predictions (e.g. fit with faces, objects, scene images, test on bodies? I apologize if I missed it in the methods).

Mismatch between methods and main manuscript for how cross-validation was done: (90/10 split, 80/20 split). Also, sometimes its written as if you fit one model on 80% and tested on 20%, as in 1 pass, versus a 5-fold cross validation (Pg 6 line 154).

Can you include information about how the ridge regression lambda parameter was set?

Statistical reporting of expert-novice comparisons could have more detail – I had a hard time linking the data plotted in Fig 3b to the stats reported (e.g. are you reporting data pooled across hemispheres here? pg 9 ln 252 e.g. unclear exactly what is being compared)

Typo: "Linear combing" pg 13 line 360

Reviewer #3 (Remarks to the Author):

This study evaluates the fit between multiple (60) deep neural network models and human hemodynamic responses to natural images in three category-selective visual areas: FFA, PPA & EBA. Four human subjects underwent extensive fMRI scanning, measuring at least 20 repetitions of each of 185 natural images. The authors employed a well-established linear encoding approach: Particular neural network layers were evaluated as potential linear bases for predicting the hemodynamic responses to held-out natural images. The prediction accuracy (i.e., the correlation between predicted and observed responses) was assessed both at the regional average activity level and at the voxel activity level. The generality of the model's predictions was tested within and across subjects. Importantly, the authors demonstrated that the best performing model explains a large portion of the variability of the responses *within* object categories, going beyond plain category-based prediction. The authors compared the accuracy of the best performing model to the judgments of human novices with respect to the category-typicality of the images, as well as to expert predictions of the potential of different images to activate the regions. Last, the authors conducted several *in silico* experiments using the fitted encoding models. They searched for activating images across large natural image sets, synthesized activating stimuli using a GAN-parameterized activation maximization procedure, and used a masking algorithm to highlight activation-driving regions in the images. The *in silico* experiments all indicated that the maximally activating stimuli for these regions are respective category-members (e.g., faces for FFA), affirming the conventional view of the category-selectivity of these regions.

Overall, this is an important contribution to the computational modeling of high-level visual responses by deep-neural-network-based models. The study reflects a considerable body of work, with multiple models and well-motivated analyses and controls. Having said this, I have several comments about the manuscript as it is, but I believe that these issues are addressable by some additional work.

Major points

1) Unaccounted-for selection bias. The authors evaluated 60 neural networks and then focused on the result from the best performing one (Resnet-50). While 10-fold cross-validation was employed to eliminate the bias introduced when fitting linear weights to map each model's activations to the brain responses, it is unclear from the methods section whether any measures were applied in order to eliminate the selection bias imposed by picking the best model among the 60 candidates. The concern is that this selection bias might cause exaggerated estimates of the prediction accuracy. Importantly, such a bias might invalidate the comparison between the best model and the human raters. Unlike the models, the accuracy of the human raters (novices and experts alike) was not subject to any maximum-taking operation, and hence the comparison might be unfair. In other words, the manuscript currently compares the best model to the average human rater, and it does so without correcting the bias incurred due to the "winner's curse".

A second source from which selection bias can creep in is the selection of the best layer in each DNN. Here as well, it is unclear whether the bias introduced by this selection operation was accounted for. To

allow for a fair assessment of the models, this bias has to be mitigated by a proper *nested* cross-validation procedure, in which the most inner fold is used to fit the encoding parameters, the intermediate fold to select the best layer (and model), and the outer fold to obtain unbiased prediction accuracy estimates. An alternative, even stronger approach is a completely held-out test dataset. Failing to do so exaggerates the reported accuracy estimate and might bias the model comparison in favor of models with more numerous layers.

2) The image set considered in the in silico image screening is very large, but it is limited to a narrow domain: natural images. This constraint means that non-natural images are never considered by this procedure. Even if one of the regions was highly responsive to non-natural, non-category-member images (for a concrete example, consider "contextually-defined faces", Cox, Meyers & Sinha, 2004 Science), this would never be discovered by the screening procedure since it tested only natural images. In principle, model-driven image synthesis (similar to that applied by Bashivan, Kar & DiCarlo, 2019 Science) could have addressed this point. However, since the current work limits the synthesized images to those produced by a strong GAN trained on natural image corpus, the stimulus-synthesis procedure's capacity to form novel, non-object-like activating images is severely limited. I suspect that if the authors had used a less restricted activation maximization procedure, less semantically-sensible activating images would have arisen, potentially revealing incorrect (i.e., adversarial) model predictions. This point touches upon a whole can of worms associated with deep neural networks as vision models (adversarial examples, metamers, and so on). The paper's discussion in its current form does not mention this obvious shortcoming of the models.

To summarize this point, the interpretation of the image screening as affirming the category-selective nature of the regions is not completely substantiated due to the limited domain of the screening images (as well as the lack of empirical testing; see the next point). The control analysis on the "face-selective" AlexNet units is necessary but not sufficient since it addresses the case of responsiveness to non-category-member natural images but not the case of responsiveness to non-natural images. Furthermore, announcing that the current approach is "subjecting claims of category selectivity to their strongest tests to date" (line 21 in the abstracts, as well as lines 266-267) neglects the extremely rich literature on modified, jumbled, or otherwise non-natural 'trick' stimuli that reveal non-trivial response properties of these high-level regions.

3) The distinctions between model predictions and empirical findings are somewhat blurred. The in silico experiments conducted on the fitted models are informative and contribute to the literature. In particular, I find Figure 4b particularly novel. However, all of the results from line 269 and onwards are model predictions rather than empirical neuroscientific findings. The authors did not call the subjects back to the scanner to test whether the natural and synthetic maximally activating stimuli predicted by the model are indeed maximally effective as stimuli for human cortices. While the authors explicitly acknowledge the distinction between predictions and results in some places in the paper (e.g., lines 377-378), this distinction is glossed over elsewhere. For example, in lines 311-312, the authors state: "This finding further strengthens the inferences that these regions are indeed selective for faces and places." This deduction does not rely on actual empirical testing.

4) The description of the data analysis procedure is insufficient for evaluation and replication. The

authors provide some external references, but these are insufficient for understanding what exactly was done. In particular, I'm concerned with the lack of description of how the best layer within each model was selected (was there nested cross-validation as suggested above?), missing details of the linear fitting (what ridge parameters were considered and how were they selected?) and absent information about the "two-stage linear mapping function". For the latter, the authors cite Bashivan, Kar & DiCarlo 2019, but it is unclear whether the implementation described there, applied to modeling particular neurons rather than voxels, is identical to what was done here.

5) Can the authors provide any form of statistical inference of the differences between the various models? The prediction accuracy (Fig. S3) seems quite similar for the top ten models. Relevant sources of variability that contribute to the ranking uncertainty are the finite sampling of stimuli, the finite (and small) sampling of subjects, and the finite number of repetitions.

6) Reliability estimates/"noise ceilings" should be plotted for each figure that depicts prediction accuracy. The way these estimates were computed should be explicitly stated in the methods section.

7) The manuscript somewhat unfairly downplays previous works on encoding visual responses by deep neural network models. In particular, Wen, Shi, Chen & Liu (2018, Scientific Reports, reference 12 in the manuscript) have used the same Resnet-50 indicated as the best model here and performed quite extensive in silico experiments on category-selectivity. It would serve the readers better to situate the current work in the context of the existing literature. Another highly relevant work (not currently cited) is Eickenberg, Gramfort, Varoquaux, and Thirion, 2017 NeuroImage.

8) Given the well-established infrastructure for sharing fMRI data (the Openneuro platform and the BIDS format) and the high potential utility of the data collected by the current study, the proposed data sharing policy ("upon request") seems to be suboptimal.

Minor points:

9) The number of participants (four) has to be mentioned in the first Results paragraph.

10) Many data points in Figure 3c have no error bars.

11) Are the images in Figures 3c and 3e randomly sampled, best performing, or manually chosen?

12) The image synthesis section does not include necessary details for replication such as optimizer choice, and optimization hyper-parameter, stopping conditions, and GAN latent initialization.

REVIEWER COMMENTS

We are very grateful to the three reviewers for their encouraging and constructive feedback on the paper, which have greatly improved the quality of the manuscript. Our responses to these comments are given below in blue, and in each case we have modified our manuscript to address these concerns (changes made to the paper are marked in red).

We also apologize for the delay in getting back with the reviewer comments caused because of COVID-related crisis in the family.

Reviewer #1 (Remarks to the Author):

The paper, "Computational models of category-selective brain regions enable high-throughput tests of selectivity", is an easy to read, clearly argued paper in which the authors present a convincing case that FFA, PPA and EBA are indeed selective for faces, places and body parts. They do this by generating encoding models that allow for the image-computability of faces, places and bodies using DNNs and BOLD-MRI data. These models have an impressive performance in their capability of predicting out sample (both stimuli and subject). Using image-computability models they show that no black swans are found (e.g. no exemplars of the wrong category) that would challenge the idea of category selectivity for places, scenes and bodies. Furthermore, images generated using these models are hyper-exemplars of their categories. The sum of these results make this a very convincing paper for the, in itself unsurprising, main point (indeed exemplar specificity). However, in doing so they authors present an approach to determine the selectivity of brain areas in a computational manner, opening up an avenue of possibilities.

However, I do have some (somewhat) more major, and minor remarks.

Somewhat more major remarks:

L200-L203, it is not clear to me if, for the voxel models, the mean ROI response (for a stimulus) is subtracted from the voxel responses. This would be necessary to make a claim on the voxel information above, and beyond, the ROI. That a voxel contains some information form the ROI is a statistical necessity that is, in itself, uninformative.

Because our scientific questions concerned inferences at the fROI-level, our main analyses were performed on voxel-averaged fROI data. This is a different approach from the main one used so far in computational neuroimaging, which has mostly considered voxel-wise or RDM-based population-level models and analyses. The voxel-wise (and RDM) analyses in our paper were performed on the raw observed responses (without removing the mean response) so that other researchers can directly compare the quality of our data and the performance of our models with other reported studies. Our main claim from this section of the paper is that the base models that are good at predicting the voxel-averaged responses are also very good at predicting the voxel-wise responses (i.e., these metrics are highly correlated).

The reviewer's suggestion raises a different but interesting question: Do individual voxels represent information distinct from the mean response? We now performed additional analyses to test this specific question. We find that there is indeed some residual variance in individual voxels even after removing the contribution of the mean response, and that our best model (ResNet50) can predict this residual variance to

some degree (see figure inset here which shows the median predictive accuracy across voxels after removing the mean responses, averaged across participants). This analysis foreshadows some of our other work in progress, which is to describe the diversity of selectivity observed within these regions (which requires further closed-loop experimental verification).

To summarize, we have now clarified the motivation for this specific analysis (see excerpt below) and leave the question of possibly distinct selectivities within individual voxels for more rigorous testing and evaluation in future work.

“The previous analyses evaluate the ability of models to predict the pooled response averaged across voxels within each category-selective fROI in an effort to build models that generalize across subjects and directly interface with the bulk of prior experimental literature on these regions. But of course information in these regions is coded by the distributed pattern of response across the neural population. So, how well do these models predict the voxel-wise responses (similar to other modeling studies^{15,16,25,31-33}) in each region?”

L218-L236, this is a fun experiment, but I am wondering if this is not very much begging the question. What does it show? The authors suggest it might indicate (I guess if novices performance match the model?) a graded version of category selectivity but I miss the necessity of this reasoning.

In this experiment, we aimed to directly compare descriptive word-based models that are usually employed to describe the function of these regions (e. g., EBA is a ‘body-selective’ region), with predictive models like DNNs. Specifically, we asked whether our computational models perform any better than the single word-level functional descriptions long used to characterize responses in these regions. One possible version of the word-level hypothesis is that the responses are binary: all body images should predict the same maximum response and all non-body should predict the same minimum response. But that hypothesis is a bit of a straw man, as it does not capture the graded nature of category membership in cognition and the clearly graded response of each region to distinct images. So, we further compared the performance of our models to a “graded” version of the category selectivity hypothesis. To do this we recruited novices with no knowledge about the FFA etc. and asked them to rate how clearly each category was depicted in each image. We then used these ratings as an operationalization of a graded version of the “word model” hypotheses of category selectivity. Specifically, we asked how well the category membership ratings could predict the observed responses in these fROIs. If novice category ratings predict the responses in each region as well as the models do, that would indicate that the models are simply capturing those “word model” intuitions. But we find that in fact the model outperforms these ratings in predicting responses to novel images, indicating that the model “knows” something beyond the graded version of the word model. We now try to more clearly motivate this analysis with an explicit question in the updated version:

“But how much better are these models than the previous descriptive characterizations used for these regions? Are the models simply capturing a graded version of category selectivity, much like human intuitions of graded category membership^{34,35}?”

L238-L261, the fun experiment continues, but basically the same question, what would it mean if experts were better, or equal, at predicting the BOLD-response to these categories than a BOLD-predicting model. For both the novel and expert I get that it showcases the superior performance of the model, but does it anything beyond this?

This analysis (like the previous one) is a way of asking what if anything the models tell us that we did not already know. The analysis above says the models have more information than the graded version of the previously used “word models” of category selectivity. Here we ask whether the

models capture anything beyond the expertise of experts in the field. The superior performance of the models compared to experts indicates that models ‘know’ something more about these regions than the experts do, that is they are not only providing image-computable versions of field expertise, but may actually be capturing something new. We use this finding as motivation to distill this additional knowledge within these models in subsequent analyses in the paper. Based on the reviewer feedback, we have now changed the introduction to this section to better motivate this question:

“The previous analysis demonstrates that the models do not merely provide image-computable versions of the word-level descriptions of the responses of each region, but embody further information not entailed in those “word models”. Here we ask whether the models “know” more about these regions than even experts in the field do.

Minor remarks

L125-L130, the vast majority of networks tested are feedforward networks, with resnet-50 being a strong ‘winner’. Given the little difference between the top performing model this is not a big point but there is an emphasis on the Cornet’s in these lines that are not justified by the tests performed, or the results discussed.

The reviewer is correct that differences between the top performing models are small. But the base-models also vary drastically in the model architecture, the specific images used to train these models (training diet), the specific task on which they were trained etc. So in this section we picked out comparisons that were more balanced and therefore justified (only within the specific comparisons). These include a discussion of few deep v/s shallow models, randomly initialized v/s trained models (where the architecture unchanged), trained on specific images (like faces) v/s trained on broad datasets (but architecture unchanged), recurrent and non-recurrent (base-architecture unchanged). It is only in this context that we report the comparison within the CORnet classes (specifically CORnetZ v/s CORnetR and S) because the CORnet models are a class of shallow models that differ in just one of these dimensions (in this case, varying recurrence while fixing the training diet, optimization goal and overall architecture) at a time.

L301-L303, how far do you need to go in the list before the first black-swan appears (as a percentage of all images tested), and what happens beyond this, a gradual transition of face-like, or a more abrupt transition (and fair enough, this is difficult to characterise)?

As Figure 4 indicates, the precise image identifier of the first black-swan (even as percentage of images) depends in large part on the composition of the stimulus-sets used for the screening procedure. But the reviewer raising an interesting question which we tried to estimate now. For this, we only considered the Imagenet dataset but even this stimulus dataset consists of ~1.3M images. So we adopted the strategy of sub-sampling 2 stimuli per 1000 images to visually inspect the images. We show the first black-swan images from the rFFA and the rEBA below (Image ID on top indicates the exact sorted ID out of 1.3M images).

For FFA we slowly observed a transition to bodies and then to round stimuli like shown below.

For the EBA we observed a transition to elongated objects and faces (which are not shown below because faces are also technically body parts). These are all very interesting predictions from the models that we plan to characterize and test in subsequent closed-loop experiments.

For PPA, we found that all the top-ranked images were clearly places/scenes, but the search for the first black swans revealed the vagueness of the verbal descriptor “place selective”; e.g., is a desktop scene, with layout similar to a room but smaller in scale, a place? This ambiguity makes

the search for PPA black-swans difficult, but in so doing nicely underscored the benefits of the computational models presented here!

L492-L494, all subjects have robust L and R EBA, FFA and PPA's. Where they selected for this, and/or is this guaranteed to happen with enough localisers?

We did not perform any subject pre-selection. In our experience, 4-5 runs of a good (face-object-scene-body) localizer (like the dynamic localizer we used in our study) are usually sufficient to robustly localize these fROIs in subjects.

Fig 2. The title is models, but only the resnet-50 results are shown. I believe the rest of the top-10 networks would be very similar but these are not shown. Is it not better to just refer to it as resnet-imagenet-50 (or an abbreviation of this)?

Our central aim in the paper is to demonstrate the success of network-based encoding models at predicting data on a fROI-level, even within categories, and even across subjects, and to demonstrate how the models can drive high-throughput experiments and stronger inferences. While our model screening efforts in this paper revealed ResNet50 to be the base-model architecture with the numerically highest prediction score, we have tested only three brain regions and a particular (small) set of natural images. As the reviewer rightly indicated the difference between the current top models is small. We therefore choose to have 'Models' in the title to make it clear that our claim is that many of these kinds of models perform well.

I really enjoyed this paper, best regards, Steven Scholte

Thank you for your encouraging comments and feedback. These suggestions have further improved the clarity of the prose and strengthened the paper.

Reviewer #2 (Remarks to the Author):

Computational models of category-selective brain regions enable high- throughput tests of selectivity

This paper measured the response properties of category-selective regions with a new high quality fMRI dataset, and then accurately modeled these responses using a deep convolutional neural network encoding-model approach. These image-computable models accounted for overall response profiles better than current intuitive models of what these regions do (with several other thoughtful tests into the different facets of responses of these regions). These models then allowed the authors to conduct a high-throughput test of the category-selective process, finding no rogue non-category selective images in a set of 3.5M would likely yield a high response in these regions. And, they probed the nature of the features encoded in these regions leveraging some of the newest advanced methods available.

The paper is clearly written; the scope of the question at hand is nicely targeted, while the methods applied to this question are broad, ranging from measures of everyday intuitions to sophisticated, cutting-edge techniques like GANs for image synthesis; the analyses are sound; the patterns of results are strong; and the claims are clear, important, and well-justified.

I also think there are a rather substantial number of patterns of data that could be further scrutinized (!) -- far beyond a single paper's worth of ideas here. As such, I am not going to tug on the theoretical threads of these datasets that I think could use more exploration (likely ideosyncratic to me), and instead will focus solely on the claims directly presented. To this end, I have only minor comments. (Relatedly, it is great that the authors are providing the data to the community for further exploration.)

Thank you for your encouraging feedback and comments which have made the paper much clearer and stronger. We too are excited to see the impact these models and data make on the community.

How good are these models? My main comment is that I was left by the end without a clear sense of the answer to whether these encoding models have completely nailed the responses, and to what degree there is representational room to grow, as measured in this fMRI dataset. For example, figure 2 a-c are rather outstanding—I think, this is pretty close to perfect. But, figure 2f it seems like there may be pretty far to go for some categories in some regions. Especially, when in figure 3c, the same data (I think?) is presented, but this time with the noise ceilings so it's easier to tell what's measurement noise vs uncaptured structure. Is there important information to be gained by the fact that e.g. the within-face predictivity is much higher than the within-body predictivity in FFA, for example?). I think the main take-home about the success of the encoding models for these regions could perhaps be clarified with some sort of synthesized summary of what they get right (which is a lot, obviously) but maybe also what your current analyses indicate that they are not capturing, given the reliable structure of responses to these 185 images.

Our within-category prediction analyses indeed expose a larger gap between the model predictions and the observed responses. As suggested by the reviewer we have now added a brief synthesized summary describing the success and the failures of the model. As the data suggest, there is a lot of across-category variance which the models (and humans) rightly pick up on, but there is also a lot of within-category variance that remains to be explained. This result also underscores the importance of collecting high-quality data which reveal the reliable differences between individual image exemplars which is necessary to expose the variance in response across these images. We now include a synthesized summary to the Results Section (see excerpt below)

“Taken together, these results show that ANN models of the ventral stream can predict the response to images in the FFA, PPA, and the EBA with very high accuracy. Further analysis on a trained ResNet50 show that the predictive accuracy of the models remain high even when tested on individual stimuli within categories and even across participants. Testing predictions within categories also exposed a larger gap between the model predictions and responses which indicates room for further model-development efforts.”

Generalizing from one subject to another subject. When reading these results I wanted a little more context: how well is the encoding model is transferring from one brain to another, given how well that brain directly correlates to the other brain? For example, is it at the noise ceiling of the data, or is there reliable structure measured in the FFA between subjects that the encoding model isn't capturing? Right now I cannot tell apart how much these analyses are telling me about the similarity and differences of people's FFAs, from the effectiveness of the encoding model itself.

The reviewer raises a fair point. We now provide the mean subject-subject agreement in Figure S4 (also shown below). This analysis shows that there is some small but reliable structure between the subjects' FFAs that the models are currently not capturing. This figure is now also included within the new Supplemental Figure 3.

Model comparison. One of the first steps was the model screening procedure, with a large number of models compared. There are some interesting models in there like “curvature” and “room-layout” that don't seem like deep net models (but maybe they are?). In any case, I wish there was a little more information/meta data about each of these models, accompanying Fig S5).

The “curvature” model is actually a deep neural network model (Taskonomy, Zamir et al., 2018) trained to extract curvature from scenes. These models (curvature, room layout etc.) are broadly known as ‘Taskonomy’ models. Each of the models are trained using a task-specific encoder-decoder network (which is the same across all tasks like extracting curvature, room-layout etc.).

The encoder network (which is a fully convolutional ResNet-50 without the pooling layer) extracts useful representations which the decoder network (a separate shallow 2-layer convolutional architecture) can use to solve these specific tasks. These models are interesting because they allow us to compare the representations extracted from the same architecture but based on different optimization goals (classification versus (say) extracting room layout etc.) and have been also used in previous comparisons between brains and neural networks.

Based on reviewer feedback, we also now include the specific citation numbers now to the Figures itself (see new Supplemental Figures 3 and 5) to direct the reader to specific paper and have added the prefix 'Tsk' to indicate that these are the Taskonomy models. The citations to the papers are also present in Supplemental Table 1.

Methods Clarifications

Can you clarify how the encoding models were fit for these finer-grained analyses for within-category predictions (e.g. fit with faces, objects, scene images, test on bodies? I apologize if I missed it in the methods).

We did not re-fit the models for the finer-grained analyses. We now explicitly mention this in the Methods.

Mismatch between methods and main manuscript for how cross-validation was done: (90/10 split, 80/20 split). Also, sometimes its written as if you fit one model on 80% and tested on 20%, as in 1 pass, versus a 5-fold cross validation (Pg 6 line 154).

Sorry about this confusion and thanks for bringing this to our attention. Our initial (voxel-wise, fROI-wise and RDM based) screening on 60 models was based on based on a 5-fold cross-validation (and repeated five times) to be computationally efficient. For the final convolutional mapping we used 10-fold cross-validation. We now repeated our initial fROI-level analyses on the 60 models based on a 10-fold cross-validation and find that it does change the rank ordering of the models (correlation between five- and ten-fold CV over 60 models: $R = 0.99$, $P < 0.000001$). We have updated the text to clarify the procedure in detail.

Can you include information about how the ridge regression lambda parameter was set?

The ridge coefficient (lambda) was set to a typical value (0.01) based on our previous work. We have now included these specifics in the Methods section.

Based on previous work on monkey ephys data, we find that sweeping through the lambda parameter within a reasonable range does not dramatically change the rank order between the models. We also confirmed this now in our fMRI data by taking a couple ridge coefficients (0.001 and 10 instead of 0.01). In both cases, changing the lambda parameter did not change the rank order between the different models (both $R = 0.99$, $P < 0.000001$)

Statistical reporting of expert-novice comparisons could have more detail – I had a hard time linking the data plotted in Fig 3b to the stats reported (e.g. are you reporting data pooled across hemispheres here? pg 9 ln 252 e.g. unclear exactly what is being compared)

The statistics were performed by comparing the model predictions with the expert/novice predictions individually on each fROI and hemisphere separately. We now specify this explicitly in the text where the statistics are reported.

Typo: “Linear combing” pg 13 line 360

Thank you! Fixed now.

Reviewer #3 (Remarks to the Author):

This study evaluates the fit between multiple (60) deep neural network models and human hemodynamic responses to natural images in three category-selective visual areas: FFA, PPA & EBA. Four human subjects underwent extensive fMRI scanning, measuring at least 20 repetitions of each of 185 natural images. The authors employed a well-established linear encoding approach: Particular neural network layers were evaluated as potential linear bases for predicting the hemodynamic responses to held-out natural images. The prediction accuracy (i.e., the correlation between predicted and observed responses) was assessed both at the regional average activity level and at the voxel activity level. The generality of the model's predictions was tested within and across subjects. Importantly, the authors demonstrated that the best performing model explains a large portion of the variability of the responses *within* object categories, going beyond plain category-based prediction. The authors compared the accuracy of the best performing model to the judgments of human novices with respect to the category-typicality of the images, as well as to expert predictions of the potential of different images to activate the regions. Last, the authors conducted several in silico experiments using the fitted encoding models. They searched for activating images across large natural image sets, synthesized activating stimuli using a GAN-parameterized activation maximization procedure, and used a masking algorithm to highlight activation-driving regions in the images. The in silico experiments all indicated that the maximally activating stimuli for these regions are respective category-members (e.g., faces for FFA), affirming the conventional view of the category-selectivity of these regions.

Overall, this is an important contribution to the computational modeling of high-level visual responses by deep-neural-network-based models. The study reflects a considerable body of work, with multiple models and well-motivated analyses and controls. Having said this, I have several comments about the manuscript as it is, but I believe that these issues are addressable by some additional work.

Thank you for your encouraging comments and constructive feedback which have substantially improved the paper. We have now performed several additional analyses and provide important clarifications which we hope address the remaining concerns.

Major points

1) **Unaccounted-for selection bias.** The authors evaluated 60 neural networks and then focused on the result from the best performing one (Resnet-50). While 10-fold cross-validation was employed to eliminate the bias introduced when fitting linear weights to map each model's activations to the brain responses, it is unclear from the methods section whether any measures were applied in order to eliminate the selection bias imposed by picking the best model among the 60 candidates. The concern is that this selection bias might cause exaggerated estimates of the prediction accuracy. Importantly, such a bias might invalidate the comparison between the best model and the human raters. Unlike the models, the accuracy of the human raters (novices and experts alike) was not subject to any maximum-taking operation, and hence the comparison might be unfair. In other words, the manuscript currently compares the best model to the average human rater, and it does so without correcting the bias incurred due to the "winner's curse".

A second source from which selection bias can creep in is the selection of the best layer in each DNN. Here as well, it is unclear whether the bias introduced by this selection operation was accounted for. To allow for a fair assessment of the models, this bias has to be mitigated by a proper *nested* cross-validation procedure, in which the most inner fold is used to fit the encoding parameters, the intermediate fold to select the best layer (and model), and the outer fold to obtain unbiased prediction accuracy estimates. An alternative, even stronger approach is a completely held-out test dataset. Failing to do so exaggerates the reported accuracy estimate and might bias the model comparison in favor of models with more numerous layers.

We thank the reviewer for raising these important points, which we address as follows:

1. Let's first consider the question of how to compare human performance to the best model without introducing a bias from the model selection procedure. As the reviewer correctly notes, the performance of each model is determined by cross validation (using identical seeds and repeated 5 times), so the predictivity values for each model is not in itself biased. The potential problem concerns the choice among 60 models, which could unfairly advantage the models over humans. We have two responses to this concern. First, the best model is actually highly constrained, as the same model had the numerically highest performance for all three fROIs. We can therefore choose the model for each fROI based on the fully independent fMRI data from the two other fROIs (for example, choose the model for FFA based on PPA and EBA). This unbiased procedure will still choose ResNet50 for all three fROIs. Second, we can give the same form of advantage to humans that we would have given to the models, by comparing the best model to the best human expert. When we do this, the best expert still performs significantly worse than the Pooled ResNet50 model ($P < 0.01$). These two points show that it is unlikely that the higher performance of models than experts is due solely to the greater degrees of freedom in choosing models than choosing humans.

2. Concerning the choice of layer, we note first that predictivity scores are based on cross-validation on held-out stimuli after the layer is chosen. Thus predictivity scores are not spuriously inflated by extra degrees of freedom in layer choice. Still, as the reviewer notes, one might nonetheless worry that base models with many layers afford a broader array of features spaces from which to choose, possibly benefitting the predictivity of networks with more layers. To test this hypothesis, we tested for a correlation across 57 models (after removing the pixel model, V1 layer of VOneNet, and the randomly initialized ResNet50 model for fairness) between the model prediction accuracy and the total number of ANN model layers. The observed correlation was close to 0 ($R = -0.17$, $P = 0.18$) indicating that models with more layers do not on average have the advantage the reviewer was concerned about.

To summarize, we are in agreement with the reviewer that it is possible for our estimates of model prediction accuracy may be inflated even when tested across participants and regions as we have done here. It is also for this reason that we feel that the model arbitration process should be done on completely new data over many regions and benchmarks. But, our models at the fROI level will make this question more easily testable by independent groups (compared to voxel-wise models). We now address this point explicitly in the discussion of the paper as well.

“Although ResNet50-V1 was the numerically most accurate base-model across regions (consistent with³¹) based on a very broad screen it is important to note that a single study or small number of regions considered (as in our own work) is insufficient to definitively determine which single base-model is the most brain-like. Ultimately the model arbitration will require a community-wide effort and rigorous model benchmarking on a larger data set with new stimuli, subjects and brain regions. An important contribution of our work is the fROI-scale of computational modeling which makes it possible to evaluate our exact model on completely independent subjects, hypotheses, and data. fROIs like the FFA, PPA, and EBA can be isolated in almost all participants and our models make testable predictions and are more directly falsifiable than say voxel-wise models (even though as we show importantly, these exact metrics are correlated across the scales of computational modeling).”

2) The image set considered in the in silico image screening is very large, but it is limited to a narrow domain: natural images. This constraint means that non-natural images are never

considered by this procedure. Even if one of the regions was highly responsive to non-natural, non-category-member images (for a concrete example, consider "contextually-defined faces", Cox, Meyers & Sinha, 2004 Science), this would never be discovered by the screening procedure since it tested only natural images. In principle, model-driven image synthesis (similar to that applied by Bashivan, Kar & DiCarlo, 2019 Science) could have addressed this point. However, since the current work limits the synthesized images to those produced by a strong GAN trained on natural image corpus, the stimulus-synthesis procedure's capacity to form novel, non-object-like activating images is severely limited. I suspect that if the authors had used a less restricted activation maximization procedure, less semantically-sensible activating images would have arisen, potentially revealing incorrect (i.e., adversarial) model predictions. This point touches upon a whole can of worms associated with deep neural networks as vision models (adversarial examples, metamers, and so on). The paper's discussion in its current form does not mention this obvious shortcoming of the models.

To summarize this point, the interpretation of the image screening as affirming the category-selective nature of the regions is not completely substantiated due to the limited domain of the screening images (as well as the lack of empirical testing; see the next point). The control analysis on the "face-selective" AlexNet units is necessary but not sufficient since it addresses the case of responsiveness to non-category-member natural images but not the case of responsiveness to non-natural images. Furthermore, announcing that the current approach is "subjecting claims of category selectivity to their strongest tests to date" (line 21 in the abstracts, as well as lines 266-267) neglects the extremely rich literature on modified, jumbled, or otherwise non-natural 'trick' stimuli that reveal non-trivial response properties of these high-level regions.

The reviewer correctly highlights a limitation of our study, which is that we only consider naturalistic stimuli (previously discussed in Lines 440-460 and made more explicit now, see below). Our claim of "subjecting category-selectivity to strongest tests yet" is with respect to the number of stimuli considered (millions in our case via our high throughput screening procedure, and perhaps much more with the GAN, compared to at best dozens in prior studies). We are currently exploring a wider space of non-natural images, but that is a much less constrained space, and will require extensive closed-loop testing, which we hope to write up in a future paper.

Based on the reviewer's suggestion we are now making the following changes:

First, we now state our claim more precisely as "subjecting category-selectivity to strongest tests yet on naturalistic stimuli". (Note that these changes have now been made in the abstract, results and the discussion accordingly)

Next, we expand on this limitation further in the Discussion:

"One route will make use of new models now being developed that include known properties of biological networks⁴²⁻⁴⁵ and that may better fit neural responses. Another route will learn from model failures in an effort to improve predictive accuracy. For example, the ANNs used here were trained on naturalistic images, are vulnerable to targeted adversarial attacks^{46,47} and are known to have limited generalizations to out-of-domain sample distributions (for instance, do not generalize from natural images to line drawings). These shortcomings suggest that these models are not likely to accurately predict observed fMRI responses to more abstract^{3,48,49} and symbolic stimuli⁵⁰⁻⁵³ including contextually-defined faces^{54,55}. Thus an important avenue of future work will entail exposing and expanding the bounds within which these models retain their predictive power. An effective strategy could be to run ever more targeted experiments that push model development efforts either by including more diverse set of stimuli for training (like sketches, line drawings, cartoons etc.), or stronger inductive biases as biological networks."

But given the reviewer's comment and the general interest in this question (and given these reviews will be published with the manuscript), we include the results of an additional analysis.

Screening on sketches instead of naturalistic images.

Here we performed our high-throughput screening procedure on stimuli from IMAGENET-Sketch by Wang et al., Neurips 2009 (downloaded from <https://github.com/HaohanWang/ImageNet-Sketch>). This database contains 50,000 images (50 images for every IMAGENET class). Note that these images are much more impoverished than the full naturalistic stimuli used in the manuscript and have very different overall image statistics. Below are the top 50 images predicted to maximally activate the FFA, PPA, and the EBA (Note that the image duplications and mirror flips are only because those images have been repeated in the image-set). These preliminary results indicate that the models still pick out images that are members of the hypothesized preferred category for these regions. We are currently in the process of expanding the stimulus domains even further. But note that these are exciting predictions that will benefit from closed-loop tests which we plan to undertake in a subsequent study on this topic.

3) The distinctions between model predictions and empirical findings are somewhat blurred. The in silico experiments conducted on the fitted models are informative and contribute to the literature. In particular, I find Figure 4b particularly novel. However, all of the results from line 269 and onwards are model predictions rather than empirical neuroscientific findings. The authors

did not call the subjects back to the scanner to test whether the natural and synthetic maximally activating stimuli predicted by the model are indeed maximally effective as stimuli for human cortices. While the authors explicitly acknowledge the distinction between predictions and results in some places in the paper (e.g., lines 377-378), this distinction is glossed over elsewhere. For example, in lines 311-312, the authors state: "This finding further strengthens the inferences that these regions are indeed selective for faces and places." This deduction does not rely on actual empirical testing.

Based on the logic of the study, we first built highly predictive encoding models of the category-selective fROIs. We then used these models to then turbo-charge our search for 'outliers' but we did not find any. Importantly, the models were in principle capable of finding such outliers (see AlexNet control). Had we found images that the models would have predicted to activate the FFA higher than faces, we would have tested them, but we did not.

To clarify, the goal of this current paper was not to demonstrate non-invasive control (for which we would have called subjects back to the scanner to check for maximum activation) but to use the models to answer the specific scientific question of category-selectivity. Despite using the high throughput screening and synthesis methods, our analyses did not reveal any violation of the prior hypothesis to test experimentally. We clarify this logic explicitly now in the text.

"Had we found any stimuli predicted to produce a strong response in a region that were not members of the hypothesized preferred category for that region, we would have cycled back to scan participants viewing those images to see if indeed they produced the predicted high responses. But we did not find such images, so there were no potentially hypothesis-falsifying stimuli to scan."

4) The description of the data analysis procedure is insufficient for evaluation and replication. The authors provide some external references, but these are insufficient for understanding what exactly was done. In particular, I'm concerned with the lack of description of how the best layer within each model was selected (was there nested cross-validation as suggested above?), missing details of the linear fitting (what ridge parameters were considered and how were they selected?) and absent information about the "two-stage linear mapping function". For the latter, the authors cite Bashivan, Kar & DiCarlo 2019, but it is unclear whether the implementation described there, applied to modeling particular neurons rather than voxels, is identical to what was done here.

We have now significantly expanded our entire Methods section with all the specific details so that readers now do not have to defer to our previous work. These details include specifically the choice of model-layer using standard cross-validation (not nested), and the ridge regression hyper-parameter used. Please also note that the two-step convolutional mapping process we used was identical to our previous work (applied here to groups of voxels instead of multi-units from monkey V4).

5) Can the authors provide any form of statistical inference of the differences between the various models? The prediction accuracy (Fig. S3) seems quite similar for the top ten models. Relevant sources of variability that contribute to the ranking uncertainty are the finite sampling of stimuli, the finite (and small) sampling of subjects, and the finite number of repetitions.

Even though technically feasible, we feel that a single study (or even a single predictive accuracy metric used here) is insufficient to make any strong inference on which particular base-model model is the most brain-like. Indeed, as our results show several ANN base-models have comparably high performance. The goal of our study here was to show how models can be used

to overcome the data-limits of current experimental methods and help us make stronger inferences on the scientific questions we explore.

We agree with the reviewer that model arbitration is an important exercise which we think will take the effort of the entire community and several experiments and benchmarking (instead of using predictive accuracy as the only benchmark) over several metrics. (We refer you to our papers on Brain-Score and integrative benchmarking for our stand on these issues)

We now expand on this in the Discussion as well:

“Although ResNet50-V1 provided the numerically most accurate models across regions (consistent with³¹) based on a very broad screen of models it is important to note that a single study or small number of regions considered (as in our own work) is insufficient to prescribe a single base-model as being the most brain-like. Ultimately the model arbitration will require a community-wide effort and rigorous integrative benchmarking on completely independent data from new subjects and evermore regions (similar to say our BrainScore platform^{14,26} for non-human primate data). But an important contribution of our work is the fROI-scale of computational modeling which makes it possible to evaluate our exact model on completely independent subjects, hypotheses, and data. fROIs like the FFA, PPA, and EBA can be isolated in almost all participants. Our models make can testable predictions and are thus more directly falsifiable than say voxel-wise models (even though as we show importantly, these exact metrics are correlated).”

6) Reliability estimates/"noise ceilings" should be plotted for each figure that depicts prediction accuracy. The way these estimates were computed should be explicitly stated in the methods section.

Based on the reviewer comments, we have added the noise-ceilings to all Figures that show predictive accuracy measures. These include Figure 2 itself (instead of just showing in Figure 3) and the Supplemental Figures. The description of how these were computed is also included in the Methods section.

7) The manuscript somewhat unfairly downplays previous works on encoding visual responses by deep neural network models. In particular, Wen, Shi, Chen & Liu (2018, Scientific Reports, reference 12 in the manuscript) have used the same Resnet-50 indicated as the best model here and performed quite extensive in silico experiments on category-selectivity. It would serve the readers better to situate the current work in the context of the existing literature. Another highly relevant work (not currently cited) is Eickenberg, Gramfort, Varoquaux, and Thirion, 2017 NeuroImage.

The Eickenberg reference was a glaring oversight which we have now included in the paper. We thank the reviewer for pointing it out. We also now emphasize the contribution and converging evidence from the Wen et al. (2018) paper, both in the Results and the Discussion.

8) Given the well-established infrastructure for sharing fMRI data (the Openneuro platform and the BIDS format) and the high potential utility of the data collected by the current study, the proposed data sharing policy ("upon request") seems to be suboptimal. The intended goal of all of our work is to compile high-quality experimental data as well as computational models and make them accessible to the community. To clarify, we will be making the models of the FFA, PPA, and EBA as well as the data used to build them publicly available on publication.

Minor points:

9) The number of participants (four) has to be mentioned in the first Results paragraph.

We have now changed the very first sentence of the Results section accordingly:

“We scanned four participants with fMRI to first localize the FFA, PPA, and EBA...”

10) Many data points in Figure 3c have no error bars.

The errorbars in Figure 3c are present, just that in some cases they are too small and occluded by the marker (circle).

11) Are the images in Figures 3c and 3e randomly sampled, best performing, or manually chosen?

Images in Figure 3c are the exact top images for the FFA and the PPA. For the EBA, we did have to remove 2 NSFW (not safe for work) images (but the 5 images included for the EBA are otherwise among the top 6 observed)

Images in Figure 3e were chosen randomly from the 10 we synthesized for each region

12) The image synthesis section does not include necessary details for replication such as optimizer choice, and optimization hyper-parameter, stopping conditions, and GAN latent initialization.

Please find the updated Methods section which describes all the specifics of the GAN synthesis procedure.

REVIEWER COMMENTS

Reviewer #1 (Remarks to the Author):

The authors have addressed the comments I had fully. I am looking forward to the next papers in this research line. Exciting new avenue.

Steven Scholte

Reviewer #2 (Remarks to the Author):

The revision has addressed my comments.

Reviewer #3 (Remarks to the Author):

The authors did considerable work improving the manuscript. In particular, I appreciate the inclusion of the noise ceiling estimates.

I am still uncertain about selection bias and cross-validation. The authors report in the response letter that "predictivity scores are based on cross-validation on held-out stimuli after the layer is chosen." This is consistent with two distinct procedures:

(I) The 5-fold cross-validation used for layer choice (line 623) and the 10-fold cross-validation used for the convolutional mapping do not use any of the "held-out 10% stimuli" (line 156). Such nesting of the model fitting cross-validation within the model evaluation cross-validation is not explicitly described in the paper.

(II) Alternatively, there is only a single level of cross-validation. Training accuracy is used for picking the best layer and for fitting the convolutional mapping. For each cross-validation fold, the unbiased accuracy of the fitted model is evaluated on the fold's test data. These accuracy estimates are then averaged across folds, pooling together evaluations of potentially distinct layer choices and convolutional mappings.

Was one of these two procedures employed? Should the reader treat all of the reported accuracies across the paper as unbiased estimates? Confusingly, the notion of holding out stimuli is introduced only in line 156 rather than at the very beginning of the methods section, suggesting that not all of the accuracy estimates were based on predicting fully held-out data.

The answers to these questions should be evident from the manuscript itself. In particular, the data handling procedure should be described in the Methods section in a level of detail that enables precise reproduction. If some of the accuracy estimates are potentially biased (i.e., they result from hyper-

parameter sweeps without external validation), this caveat should be transparently reported and discussed.

As mentioned in the previous review, this concern relates not only to the numerical estimates themselves but also to the validity of the comparison between model and human predictions (Figure 3).

REVIEWER COMMENTS

We are happy to note that our previous submission addressed the vast majority of reviewers' comments. We have now clarified the one remaining concern raised by Reviewer 3. Our responses to these comments are given below in blue, and in each case we have modified our manuscript to address these concerns (changes made to the paper are marked in red).

Reviewer #1 (Remarks to the Author):

The authors have addressed the comments I had fully. I am looking forward to the next papers in this research line. Exciting new avenue.

Steven Scholte

We thank the reviewer for their valuable comments and feedback on the paper.

Reviewer #2 (Remarks to the Author)

The revision has addressed my comments.

We thank the reviewer for their valuable comments and feedback on the paper.

Reviewer #3 (Remarks to the Author)

The authors did considerable work improving the manuscript. In particular, I appreciate the inclusion of the noise ceiling estimates.

I am still uncertain about selection bias and cross-validation. The authors report in the response letter that "predictivity scores are based on cross-validation on held-out stimuli after the layer is chosen." This is consistent with two distinct procedures:

(I) The 5-fold cross-validation used for layer choice (line 623) and the 10-fold cross-validation used for the convolutional mapping do not use any of the "held-out 10% stimuli" (line 156). Such nesting of the model fitting cross-validation within the model evaluation cross-validation is not explicitly described in the paper.

(II) Alternatively, there is only a single level of cross-validation. Training accuracy is used for picking the best layer and for fitting the convolutional mapping. For each cross-validation fold, the unbiased accuracy of the fitted model is evaluated on the fold's test data. These accuracy estimates are then averaged across folds, pooling together evaluations of potentially distinct layer choices and convolutional mappings.

Was one of these two procedures employed? Should the reader treat all of the reported accuracies across the paper as unbiased estimates? Confusingly, the notion of holding out stimuli is introduced only in line 156 rather than at the very beginning of the methods section, suggesting that not all of the accuracy estimates were based on predicting fully held-out data.

The answers to these questions should be evident from the manuscript itself. In particular, the data handling procedure should be described in the Methods section in a level of detail that enables precise reproduction. If some of the accuracy estimates are potentially biased (i.e., they result from hyper-parameter sweeps without external validation), this caveat should be transparently reported and discussed.

As mentioned in the previous review, this concern relates not only to the numerical estimates themselves but also to the validity of the comparison between model and human predictions (Figure 3).

Our cross-validation procedure is neither of those described by the reviewer. All prediction scores and accuracies reported in the paper are based on completely independent data that were never used to select the network or layer or to build the model (mappings). Here is how we accomplished this. To choose the network and layer for each fROI, we used the completely independent data from the homologous region in the opposite hemisphere. For example, the network and layer used for the right FFA model was chosen as the network and layer that yielded the highest accuracy in the left FFA and vice versa. Once the network and layer were thus chosen for each fROI, the model was built from one subset of the data and the accuracy of that model was determined from completely held-out data (stimuli). Thus all accuracies reported in the paper are unbiased, and the comparison of model accuracies to humans does not give an unfair advantage to models. This procedure is now made clearer in the paper.

To ensure that the readers do not have a similar concern of data non-independence we include the following sentences under the section on ‘Choice of base-model and best model layer’ in the Methods.

“Note that the best layer for a given fROI (for example, the left FFA) was determined based on independent data from the homologous region in the other hemisphere (the right FFA in this example) to ensure that all the parameter estimates are fixed before determining the cross-validated prediction accuracies.”

Please also note that cross-validation and data independence was in fact introduced in the first results paragraph itself. We have now tweaked the sentence further to draw the readers’ attention to the appropriate section on data independence and encoding models in the Methods directly from the manuscript as the reviewer suggests.

“We then tested the accuracy of this model at predicting responses to completely held-out stimuli (aka. cross-validation, scored as the Pearson correlation of the predicted vs. the observed responses on those held-out of stimuli, see section on Encoding Models in Methods”

We have also now added this additional sentence where we first introduce encoding models in the Methods.

“Note that all the predictions reported in the paper are always based on completely independent data not used to train the model and with the specific layer choice determined using completely independent neural data from the homologous regions in the other hemisphere (see below).”

And finally, we have completely rewritten the entire section on “1. Choice of base-model and best model layer” under Methods, to clearly describe how these choices were determined.

We thank the reviewer again for raising this important point and hope this additional clarification and additions to the manuscript clarify the issue and allay any remaining concerns.

REVIEWER COMMENTS

Reviewer #3 (Remarks to the Author):

Hemodynamics responses to the same stimulus set measured in homologous regions (e.g., right FFA and left FFA) within a subject are far from being independent data. Consider the solid interhemispheric correlations between homologous visual regions reported by Davies-Thompson and Andrews (2012, Journal of Neurophysiology). Therefore, while the usage of homologous regions might reduce the problem of "double-dipping", it does not eliminate it. Since the selection of the base model and its layer seems to be informed by the entire stimulus set (i.e., 185 images), subsequent analyses that split these 185 images again into cross-validation folds do not test "completely held-out stimuli" (line 113), and hence they cannot yield unbiased accuracy estimates.

An additional potential source for a positive bias is the 6x6 grid search for the mapping function's hyper-parameters. The subsection "Fitting the fROI responses from activations in the best layer" in its current form does not preclude the possibility that this cross-validated grid-search used all of the 185 images, which were again split for the third time in order to perform the final cross-validation. This is my best understanding of the text as it is. Since there is no explicit description in the "Fitting the fROI..." subsection (or elsewhere) of neither the hyper-parameter tuning cross-validation nor the final cross-validation, it is hard to judge how the data were handled.

I am not implying that this potential bias is necessarily large, but its existence cannot be precluded given the available description of the data analysis procedures. While this problem does not undermine most of the manuscript's results, it further reduces my confidence in the validity of the comparison between the neural network model and the human judgments (for which no mapping parameters were estimated).

In fact, if we seek to compare the human judgments with the neural network model predictions in a fairer manner, we should probably use the model accuracy estimates that were obtained by cross-validation across subjects (Figure S4, third column from the left). The human ratings should not be compared to within-subject model accuracy estimates since the human raters had no access to subject-specific feature weighting. In other words, the human raters did not peek into the fMRI recordings and finetuned their predictions to individual subjects' idiosyncratic response patterns. Judging by the fairer (and seemingly, less biased) measure of model accuracy cross-validated across subjects, it appears that the experts are *slightly better than the models* (0.84 ± 0.01 for the experts, line 257 vs. 0.82 ± 0.01 for the cross-subject cross-validated model accuracy estimates, line 151).

I hate to delay a publication in an advanced stage. Still, my understanding is that this manuscript requires the following adjustments: (a) a fully transparent description of data handling: which stimuli were used in each of the three cross-validation procedures mentioned in the paper, and in what sense stimuli were held out in the final testing. This would enable the reader to judge the procedure and replicate it (b) removal of the recently added claims about "completely independent data" which do not seem to be warranted (c) acknowledging potential sources of bias and their implication on the interpretation of the results (d) using cross-subject instead of within-subject cross-validation when

comparing the neural network model to human experts and novices. This latter change might entail letting go of the finding of an apparent superiority of the models over the experts.

Minor points:

1. there is a typo or an omitted word in line 640 ("Note that this choice is highly constrained does not change even when").
2. The references to Figure S3 in lines 158 and 163 are probably meant to relate Figure S4.

REVIEWER COMMENTS

Here we fully address further concerns raised by the reviewer. We also now include the new analysis requested by the reviewer (which we had in fact already performed and mentioned in the previous version of the paper). This analysis does not importantly change any of the results. Our responses to the reviewer's comments are given below in blue (changes made to the paper are marked in red).

Reviewer #3 (Remarks to the Author):

Hemodynamics responses to the same stimulus set measured in homologous regions (e.g., right FFA and left FFA) within a subject are far from being independent data. Consider the solid interhemispheric correlations between homologous visual regions reported by Davies-Thompson and Andrews (2012, Journal of Neurophysiology). Therefore, while the usage of homologous regions might reduce the problem of "double-dipping", it does not eliminate it. Since the selection of the base model and its layer seems to be informed by the entire stimulus set (i.e., 185 images), subsequent analyses that split these 185 images again into cross-validation folds do not test "completely held-out stimuli" (line 113), and hence they cannot yield unbiased accuracy estimates.

An additional potential source for a positive bias is the 6x6 grid search for the mapping function's hyper-parameters. The subsection "Fitting the fROI responses from activations in the best layer" in its current form does not preclude the possibility that this cross-validated grid-search used all of the 185 images, which were again split for the third time in order to perform the final cross-validation. This is my best understanding of the text as it is. Since there is no explicit description in the "Fitting the fROI..." subsection (or elsewhere) of neither the hyper-parameter tuning cross-validation nor the final cross-validation, it is hard to judge how the data were handled.

I am not implying that this potential bias is necessarily large, but its existence cannot be precluded given the available description of the data analysis procedures. While this problem does not undermine most of the manuscript's results, it further reduces my confidence in the validity of the comparison between the neural network model and the human judgments (for which no mapping parameters were estimated).

In fact, if we seek to compare the human judgments with the neural network model predictions in a fairer manner, we should probably use the model accuracy estimates that were obtained by cross-validation across subjects (Figure S4, third column from the left). The human ratings should not be compared to within-subject model accuracy estimates since the human raters had no access to subject-specific feature weighting. In other words, the human raters did not peek into the fMRI recordings and finetuned their predictions to individual subjects' idiosyncratic response patterns. Judging by the fairer (and seemingly, less biased) measure of model accuracy cross-validated across subjects, it appears that the experts are *slightly better than the models* (0.84 ± 0.01 for the experts, line 257 vs. 0.82 ± 0.01 for the across-subject cross-validated model accuracy estimates, line 151).

I hate to delay a publication in an advanced stage. Still, my understanding is that this manuscript requires the following adjustments:

(a) a fully transparent description of data handling: which stimuli were used in each of the three cross-validation procedures mentioned in the paper, and in what sense stimuli were held out in the final testing. This would enable the reader to judge the procedure and replicate it.

Based on the reviewer's suggestion we have now even more fully fleshed out the entire analysis and data handling procedure. Briefly, all the choices (which model layer of the base-model, *as well as the* convolutional mapping hyperparameters) for a given fROI, were obtained always based on distinct neural data from the opposite hemisphere (though indeed based on the same 185 images). The analysis pipeline is now described in great detail and depicted schematically in the new **Fig. S13** and accompanying text.

We have added the following information in the Methods:

“The convolutional mapping has two hyperparameters that control the degree of weight regularization for the spatial and encoding dimensions. We considered 6 points, spaced evenly on a logarithmic scale, within the [0.01 100] range. We chose these hyperparameters by performing a grid-search on these two hyperparameters using a separate 10-fold cross-validation procedure. As before, the grid-search was performed on distinct neural data from the homologous region in the opposite hemisphere (on the same set of 185 images) and on the model-layer determined from 1) above. The final model prediction accuracy for the fROI was then determined (using features from the specific model-layer, and using grid-search regularization weights determined from homologous regions from the opposite hemisphere) using a single 10-fold cross-validation (using a different random seed for both weight initialization and data split from before) over the 185 images. This entire procedure is outlined schematically in **Fig. S13a**.”

In addition, we also now include a new **Fig. S13**. This figure schematically outlines the data handling procedures.

(b) removal of the recently added claims about "completely independent data" which do not seem to be warranted.

We agree and have now replaced “completely independent data” with “distinct neural data”.

(c) acknowledging potential sources of bias and their implication on the interpretation of the results

The reviewer is technically correct in pointing out that data from across hemispheres may not be fully independent because of say shared noise etc. We also agree that these effects, if any, are likely small and mainly affect the comparison between models and experts.

We now bring this point up in both the **Methods** and the **Results** section of the paper:

Methods:

We have added the following paragraph to the end of the section where we describe how the final prediction accuracies were computed.

“Note that we always used distinct neural data to determine all the free parameters (choices on which base-model, layer and grid-search hyperparameters) to reduce any model selection bias. Nonetheless it may be argued that neural data across hemispheres and regions are not entirely independent because of perhaps shared scanner noise even across hemispheres within each individual fMRI participants’ data (see ⁶⁹ for instance). While we cannot entirely rule out the influence of this shared noise on the reported neural predictivity estimates, these effects, if any, are likely small and do not qualitatively affect the majority of results. For quantitative comparison of model prediction accuracy to humans, we avoid this problem by crossvalidating across both images and participants (see section on Behavioral experiments below).”

We also refer to this issue again when we discuss the new analysis comparing models with humans to draw the readers’ attention to this possible issue before describing the updated analyses

“In order to fully allay any concern about prediction accuracies being biased in favor of models (see last paragraph on Encoding Models above), we used models cross-validated across both participants and images (similar to Fig. S4, third column from left).”

Results:

Please see the updated section of the revised paper comparing models with experts and humans. We have completely redone this analysis as suggested by the reviewer, now cross-validating across both stimuli and subjects, so that the accuracies are now completely independent of the data used to nail down all the parameters. The results, which now appear in the updated Figure 3, are qualitatively unchanged. These additions are further expanded in (d) below. As requested by the reviewer, the additions there describe the potential bias in comparing models with humans and how we avoided them.

This includes the following addition to the introduction to this section:

“A challenge in this analysis is that model prediction accuracies might be slightly overestimated because the distinct data used to lock down model parameters (from the opposite-hemisphere fROI) might not be fully independent because of correlated noise between homologous regions in opposite hemispheres. To avoid this potential problem, we tested the ability of models and humans to predict each fMRI participant's individual held-out data (instead of the pooled data averaged across participants, see section on Behavioral experiments under Methods). That is, for these analyses we used models not trained on any of the specific to-be-predicted subjects' data (or images), paralleling the situation of our human participants' predictions.”

(d) using cross-subject instead of within-subject cross-validation when comparing the neural network model to human experts and novices. This latter change might entail letting go of the finding of an apparent superiority of the models over the experts.

We agree. In fact, we had already performed this exact analysis and referred to it at the end of the section comparing models with humans (excerpt quoted below from the previous draft of the manuscript):

“We observe qualitatively similar results even when we performed a more stringent analysis based on building models based on N-1 subjects and evaluated predictions for the Nth subject (thus cross-validating not only across images but also across subjects).”

Given the reviewer's suggestion, we now present that analysis as the main analysis comparing humans to models. We have thus entirely updated Figure 3, which now shows only the results of the analysis where we crossvalidate across both subjects and images. This updated analysis is now based on evaluating the ability of models and humans to predict the neural data for each individual fMRI participant (instead of predicting the pooled data across participants, as before), and based on cross-validating over *both* participants and images. Importantly all the major findings remain. The reason this analysis works (when the reviewer predicted it would not) is that instead of predicting the *pooled* response across subjects (which has higher SNR because of averaging), the models and hence also humans (novices and experts) are now required to predict the observed response of individual held-out fMRI subjects. Individual-subject fMRI data are noisier than pooled data across participants, lowering the prediction accuracy for both models and humans (mean accuracy across fROIs is now 0.82 ± 0.01 for models, 0.77 ± 0.01 for experts and 0.64 ± 0.04 for the novices, with models remaining significantly more accurate than experts ($P = 0.03$, Wilcoxon signed-rank test) as before.

We have also incorporated the following changes to the paper:

1. We have completely updated the Results section describing the comparisons between models and humans (“ANN models of the ventral stream have higher predictive accuracy than category-based descriptive models and domain experts”).

This includes the following note to draw the attention of the readers to the issue of comparing models with humans.

“A challenge in this analysis is that model prediction accuracies might be slightly overestimated because the distinct data used to lock down model parameters (from the opposite-hemisphere fROI) might not be fully independent because of correlated noise between homologous regions in opposite hemispheres. To avoid this potential problem, we tested the ability of models and humans to predict each fMRI participant's individual held-out data (instead of the pooled data averaged across participants, see section on Behavioral experiments under Methods). That is, for these analyses we used models not trained on any of the specific to-be-predicted subjects' data (or images), paralleling the situation of our human participants' predictions.”

2. Figure 3 has also been updated and now shows the fully cross-validated data (across images and participants).

3. The Methods have been changed to describe how the match to observed data is now computed for humans (experts and novices)

4. The Methods section under “Behavioral experiments” now includes an extra paragraph that describes the challenges in comparing models and humans on the same set of 185 images, and our solution to that challenge:

“In order to fully allay any concern about prediction accuracies being biased in favor of models (see last paragraph on Encoding Models above), for the comparison of models to humans we used models cross-

validated across both participants and images (similar to Fig. S4, third column from left). Thus, for predicting the fMRI responses in each subject, we used computational models trained using a 10-fold cross-validation on all other subjects. This procedure is outlined schematically in **Fig. S13b**. In this way the models did not use any of that particular Nth fMRI participants' individual data (nor those specific images on which the predictions are made) in the model training procedure."

Minor points:

1. there is a typo or an omitted word in line 640 ("Note that this choice is highly constrained does not change even when").

Fixed

2. The references to Figure S3 in lines 158 and 163 are probably meant to relate Figure S4.

Thanks. We have fixed it now.

We hope the reviewer is happy with these comprehensive changes which address all of the reviewer's comments.

REVIEWERS' COMMENTS

Reviewer #3 (Remarks to the Author):

The authors have addressed my concerns.